# Fueling sentinel node via reshaping cytotoxic T lymphocytes with a flex-patch for post-operative immuno-adjuvant therapy

Bei Li[1,2,3], Guohao Wang[1,2,3], Kai Miao [1,2,3], Aiping Zhang[1,2], Liangyu Sun [1], Xinwang Yu[1,2], Josh Haipeng Lei[1,2], Lisi Xie[1,2], Jie Yan[1,2], Wenxi Li[1,2], Chu-Xia Deng[1,2] ✉ & Yunlu Dai [1,2] ✉

Clinical updates suggest conserving metastatic sentinel lymph nodes (SLNs) of breast cancer (BC) patients during surgery; however, the immunoadjuvant potential of this strategy is unknown. Here we leverage an immune-fueling flex-patch to animate metastatic SLNs with personalized antitumor immunity. The flex-patch is implanted on the postoperative wound and spatiotemporally releases immunotherapeutic anti-PD-1 antibodies (aPD-1) and adjuvants (magnesium iron-layered double hydroxide, LDH) into the SLN. Genes associated with citric acid cycle and oxidative phosphorylation are enriched in activated CD8$^+$ T cells (CTLs) from metastatic SLNs. Delivered aPD-1 and LDH confer CTLs with upregulated glycolytic activity, promoting CTL activation and cytotoxic killing via metal cation-mediated shaping. Ultimately, CTLs in patch-driven metastatic SLNs could long-termly maintain tumor antigen-specific memory, protecting against high-incidence BC recurrence in female mice. This study indicates a clinical value of metastatic SLN in immunoadjuvant therapy.

Surgical treatment of breast cancer (BC) generally involves the dissection of sentinel lymph nodes (SLNs) where tumor cells are detected[1,2]. However, clinical follow-up studies show that for BC patients with a moderate tumor burden in SLN, no axillary dissection does not show inferiority in disease-free survival compared with axillary dissection[3,4]. Given the much higher risks of surgical complications and more exhaustive histological examinations in axillary dissection over no axillary dissection, the former is also suggested as an overtreatment[5]. In line with this perspective, SLN, as the first leukocyte niche to encounter tumor antigens from the primary tumor, enables the immune response to orchestrate spatiotemporally, which is critical for tumor vaccination[1,6]. Collectively, conserving metastatic SLN during BC surgery is a noninferior strategy; however, whether it could be utilized for postsurgical adjuvant therapy, protecting against high-incidence BC resurgence[7], is still unknown.

An emerging notion is that SLN lies at the crossroads of tumor-promoting metastasis and antitumor immunity[8]. A peripheral, primary tumor interconnects SLN via a network of lymphatic vasculatures where the former can suppress SLN progressively to be a permissive environment for tumor cell invasion[1,9–11]. Representatively, tumor expression of vascular endothelial growth factor C promotes deletional tolerance and dysfunction of tumor-specific CD8$^+$ T cells[11,12]. CD4$^+$ T cells within metastatic SLNs fail to efficiently present antigens and exhibit the expansion of regulatory T (Treg) cells[13,14]. However, these immunosuppressive factors do not render SLN antitumor therapy impractical. Removing tumor-draining lymph nodes apparently impedes tumor regression induced by the checkpoint administration of programmed cell death 1 (PD-1)[15] and decreases intratumoral infiltration of effective tumor-specific CD8$^+$ T cells[16]. When monitoring tumor patients positively responding to PD-1 blockade therapy,

[1]Cancer Centre and Institute of Translational Medicine, Faculty of Health Sciences, University of Macau, Macau SAR 999078, China. [2]MoE Frontiers Science Center for Precision Oncology, University of Macau, Macau SAR 999078, China. [3]These authors contributed equally: Bei Li, Guohao Wang, Kai Miao. ✉e-mail: cxdeng@um.edu.mo; yldai@um.edu.mo

phenotypic changes of exhausted CD8[+] T cells in peripheral blood, but not in their intratumoral area, have been identified[17,18]. Anti-PD-1 antibodies (aPD-1) appear to modulate the protumoral tendency of SLN, and the educated SLN endows aPD-1 with improved antitumor effectiveness in return.

In addition, we speculate that functional immune adjuvants regulate the activation and cytotoxicity of CD8[+] T cells in SLNs. Of note, $Mg^{2+}$ ions improve the glycolytic metabolism and cancer-directed effector activity of effector CD8[+] T cells via T-cell receptor (TCR) signal strengthening[19]. Rather than being metabolic waste that can oxidize cellular components, reactive oxygen species (ROS) have been reported in numerous studies to show a beneficial role in T-cell activation[20,21]. They can enhance TCR signaling by regulating redox-related proteins that participate in T-cell activation[20]. Introducing $Mg^{2+}$ and the ROS-promoter $Fe^{3+}$ into an immune adjuvant may synergistically influence the functional activities of CD8[+] T cells and modulate their immunoresponsive outcomes. As a typical immunoadjuvant nanosheet, layered double hydroxide (LDH) has a sandwich framework comprising edge-sharing $M^{2+}(OH)_6$ and $M^{3+}(OH)_6$ octahedral units ($z = 1$ or $2$; M, metal cations) on the hydroxide host layer and intercalating exchangeable anions between the layers[22,23]. This unique

structure makes LDH a superior adjuvant candidate for integrating functional $Mg^{2+}$ and $Fe^{3+}$ ions for efficient SLN immune regulation.

Here, we show a postsurgical flex-patch that electrostatically coentraps functional aPD-1 and $Mg^{2+}Fe^{3+}$-based LDH (labeled as LDH in the following context) via biocompatible carboxylated chitosan (CC) to fuel SLN for postsurgical BC adjuvant therapy. The flex-patch, after being implanted on the wound bed after primary tumor removal, exhibits spatiotemporal and sustained SLN delivery (Fig. 1), facilitating early therapeutic intervention in patients. Regarding SLN immunomodulation driven by the patch, delivered aPD-1 and LDH improve the glycolytic activity of activated CD8[+] T cells (CTLs) suppressed by invaded tumor cells. LDH boosts CTL activation and cytotoxic killing via metal cation ($Mg^{2+}$ and $Fe^{3+}$)-mediated shaping. A combination of aPD-1 and LDH adjuvant augments tumor-specific CTLs in metastatic SLN, a tumor-mediated immune niche, and triggers memory CTL generation and phenotypic differentiation. According to investigations in two different BC models (4T1 and EMT6), fueling SLN with the flex-patch inhibits postoperative tumor recurrence by over 80%, and the cured ones exhibit robust resistance to tumor rechallenge. This study provides an insight into whether metastatic SLNs could be leveraged for postsurgical clinical care.

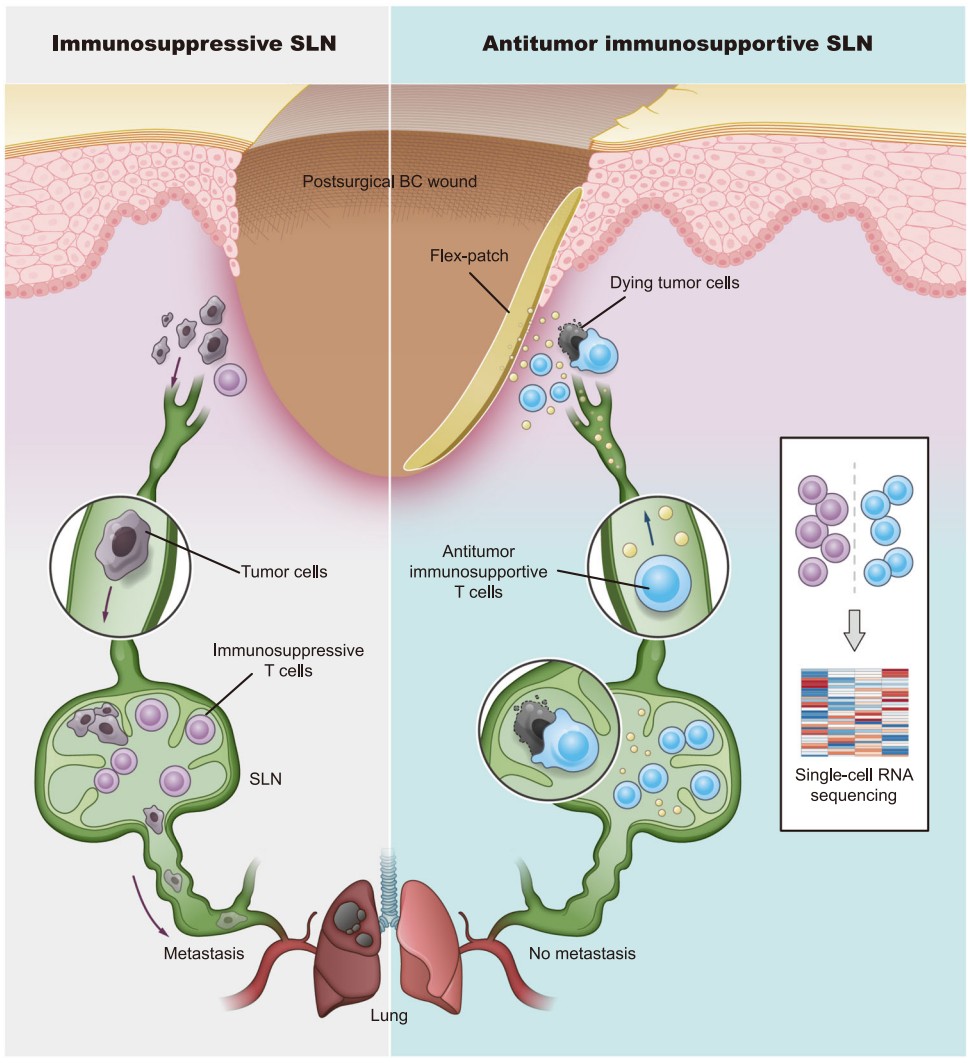

**Fig. 1 | Schematic of an implantable flex-patch to reinvigorate SLN for postsurgical immunotherapy.** In a postsurgical BC case with a conserved metastatic SLN, immune agents released from the flex-patch adhesive to the surgical wound bed travel into the immunosuppressive SLN spatiotemporally and reshape the SLN for efficient antitumor immunity. To comprehensively profile T lymphocytes in both immunosuppressive and immunosupportive states, a single-cell RNA sequencing technique was applied.

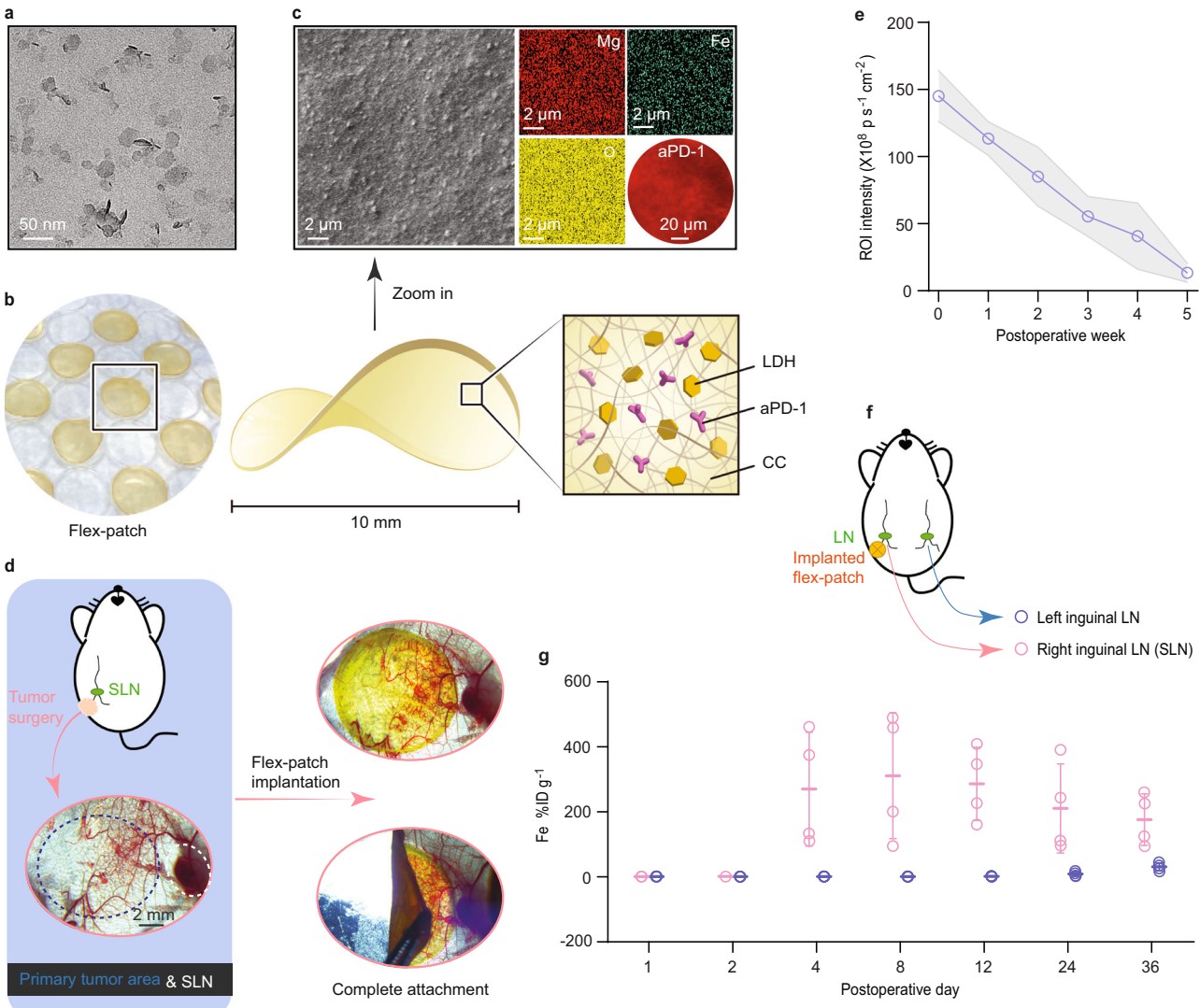

**Fig. 2 | Characterization and sustained SLN accumulation of the flex-patch.**
**a** Representative transmission electron microscopy (TEM) image of LDH nanosheets (scale bar: 50 nm). **b** Photograph and schematic contents of the flex-patch.
**c** Representative scanning electron microscopy (SEM) image of the flex-patch with elemental mapping (scale bar: 2 μm). Uniform distribution of aPD-1 labeled with PE fluorophore was imaged by fluorescence microscopy (scale bar: 20 μm).
**d** Microscopic images showing the implantation of the flex-patch after primary tumor surgery (scale bar: 2 mm). **e** Fluorescence quantification of IVIS imaging

depicting the sustained retention and disassembly of the implanted patch in vivo. $n = 3$ mice. **f** Schematic illustrating the experimental details in (**g**). **g** Quantitative analysis delineating patch accumulation in the SLN and its contralateral LN. Concentrations of released items were normalized as the percentage of accumulated dose of Fe per gram of LN ($\%ID\ g^{-1}$). $n = 4$ mice at each time point per group. All experiments for imaging collection were repeated three times. For (**e**) and (**g**), data are presented as the mean ± SD. Source data are provided as a Source Data file.

## Results

### Flex-patch features efficient SLN delivery

LDH nanoadjuvants were prepared by coprecipitating magnesium nitrate and iron(III) nitrate (Supplementary Fig. 1a), as detailed in the "Methods" section. These sheet-like nanoproducts (Fig. 2a) were well dispersed with a mean particle size of 40 nm (Supplementary Fig. 1b) and exhibited a powder X-ray diffraction (XRD) pattern typical of the LDH phase[22], which is indexed to the (003), (006), and (110) diffractions in Supplementary Fig. 1c. The presence of iron(III) in LDH induced a high positive charge (30.7 ± 1.1 mV, Supplementary Fig. 1d). Biocompatible CC[24] was adopted as a matrix to package functional aPD-1 and LDH adjuvants given its enriched carboxylate groups. Following an electrostatic interaction between amino-rich aPD-1, positively charged LDH, and negatively charged CC (Supplementary Fig. 1d), we assembled these three functional components via simple ultrasonic mixing (Supplementary Fig. 1a), detailed in "Methods". Desiccating the mixture on the cell culture lid gently shaped the final product into a round

flex-patch with a 10-mm diameter (Fig. 2b). Zooming in on the patch with scanning electron microscopy (SEM) and fluorescence microscopy, we observed a rough surface of the patch where LDH (mapped by Mg and Fe elements) and aPD-1 (labeled with fluorophore PE) were uniformly distributed (Fig. 2c).

Immersing the flex-patch into phosphate buffer with a pH of 6.5 (to mimic an acidic wound environment[25]), we confirmed that co-entrapped aPD-1 and LDH nanoparticles could be disassembled slowly from the patch (Supplementary Fig. 1c–g). The released nanoparticles retained their size distribution as the preassembly (Supplementary Fig. 1b, f), and kept the LDH framework, supported by their LDH crystal structure ((003), (006), and (110) peaks in Supplementary Fig. 1c) and FTIR spectra (Supplementary Fig. 1h) at 800–480 cm$^{-1}$ (vibrations of M–O (M = Mg, Fe) units of LDH) and 3800–3000 cm$^{-1}$ (vibrations of O–H groups of LDH). Weak carboxylate spectra at ~1550 and 1440–1340 cm$^{-1}$ were also detected, and they were attributed to surface-adsorbed CC. The adsorbates shifted LDH with a negative charge

($-28.3 \pm 2.1$ mV, Supplementary Fig. 1d), which may benefit the body fluid migration of these released particles[26,27].

In the above disassembling test (Supplementary Fig. 1e), we also noticed that the buffer pH value increased to ~7.9, which may be correlated with the acid neutralization of alkaline Mg–O(H) groups in LDH. Treating LDH nanosheets with plenty of acidic buffer revealed that $Mg^{2+}$ ions were first etched off from the LDH skeleton, while $Fe^{3+}$ was not (Supplementary Fig. 1i); however, adequate $Mg^{2+}$ decomposition from LDH appeared to sequentially dismiss $Fe^{3+}$. This is due to a significant difference in the solubility product constants (Ksp) of magnesium hydroxide (Ksp: $5.61 \times 10^{-12}$) and iron(III) hydroxide (Ksp: $2.79 \times 10^{-39}$)[28]. The pH(6.5)-responsive Mg–O(H) groups contributed an extra acid neutralization performance of the LDH and guaranteed the complete biodegradability of LDH, as no Fe-Fe close contacts[29] existed in our LDH (with a 4:1 molar ratio of Mg to Fe), as indicated in Supplementary Fig. 1j.

To investigate the in vivo applications of the flex-patch, an orthotopic 4T1 breast tumor model with high spontaneous metastasis was established. 4T1 cancer cells were labeled with GFP and luciferase (4T1-GFP-luc) to facilitate tracking. On day 10 postinoculation, when the primary 4T1 tumor grew to ~500 mm³, we collected brachial LNs, axillary LNs, and inguinal LNs from the mouse model and identified luminescent SLNs (Supplementary Fig. 2a, b). When fixing the SLN for immunohistochemistry (IHC) analysis, strong 4T1-related GFP expression (Supplementary Fig. 2c) was observed. Both lines of evidence reveal clear SLN invasion of 4T1 cancer cells. This tumor infiltration upregulated cellular PD-1 level in SLNs (Supplementary Fig. 3), including that of activated CD8+ T cells (Supplementary Fig. 4).

Tumor surgery was thus conducted on day 10 after 4T1 inoculation and conserved the metastatic SLN. The flex-patch was then adhered to the wound bed with good affinity (Fig. 2d). To track patch disassembly and accumulation in SLNs, we inserted fluorophore Cy5.5 into the LDH interlayer. Successful intercalation enlarged the interlayer space and led to a left shift of the (00l) peaks of LDH (l = 3 or 6; Supplementary Fig. 1c)[30,31]. From fluorescence IVIS imaging (Fig. 2e and Supplementary Fig. 5), we observed a 5-week sustained disintegration of the patch. Those released agents gradually became concentrated in SLN from the postoperative 4th day and peaked on the 8th day, as demonstrated by LDH-elemental (i.e., Fe) quantifications in SLN (Fig. 2f, g). When prolonging the monitoring duration, we found that only a slightly higher Fe concentration was tested in other main organs (Supplementary Fig. 6) compared with the Fe %ID per SLN gram. It guaranteed no obvious histopathological variation in major organs (Supplementary Fig. 7a) during patch treatment. Upon further analysis of the biochemical markers in murine sera, the implanted patch barely affected the levels of blood urea nitrogen (BUN), creatine kinase (CK), lactate dehydrogenase (LADH), creatinine (CRE), aspartate transaminase (AST), and alanine aminotransferase (ALT) (Supplementary Fig. 7b). Therefore, as confirmed by good cellular safety (Supplementary Fig. 8), our flex-patch also exhibits good biocompatibility and biosafety for in vivo applications.

## aPD-1 and LDH modulate the metabolic activities of activated CD8+ T cells in SLN

As dynamic cells, the fate and immune function of T cells are intimately linked to their metabolic programming[32]. Here, from the collected SLNs, activated CD8+ T cells were identified by scRNA seq to explore their transcription-profiled immunometabolism with and without aPD-1 and LDH treatment. The same surgical procedure and patch implantation were conducted first (Fig. 3a). On day 10 postimplantation, SLNs for each group were harvested and dissociated for scRNA seq. As elucidated in the uniform manifold approximation and projection (UMAP) plot, unsupervised clustering identified five types of cells (T cells, B cells, plasma B cells, myeloid cells, and natural killer

cells) and three T phenotypes (CD8+, CD4+ and CCR6+ T cells) in SLN based on their distinct marker genes (Supplementary Fig. 9).

In the identified CD8+ T subsets (Supplementary Fig. 10a), activated CD8+ T cells were highlighted (Fig. 3b) to investigate their differentially expressed genes (DEGs). Although without proportional variation compared with CC treatment (Supplementary Fig. 10b), these immune cells after aPD-1@CC therapy (Fig. 3c) exhibited the upregulation of TCR signaling (VPS37B), effector function (IFRD1), and survival (HSPA5, MAP1LC3B)[33,34]. Similar improvements (Fig. 3d) involving TCR signaling (VPS37B), activation (AMD1 and IRS2), effector function (IFRD1), proliferation (DUSP10), and persistence (FOSL2)[35–38] were also pronounced after LDH@CC management with an increased proportion of activated CD8+ T cells (Supplementary Fig. 10b). When gene set enrichment analysis (GSEA) was further conducted on DEGs, in immunosuppressive SLNs (CC, Fig. 3e, f), the mitochondrial activities of activated CD8+ T cells featured three important pathways: the citric acid cycle (also known as the TCA cycle), respiratory electron transport, and oxidative phosphorylation (also known as OXPHOS). However, none of these pathways were enriched after aPD-1@CC or LDH@CC administration. According to pioneering studies, the TCA cycle and OXPHOS pathways indicate cellular quiescence with basal metabolic demand and are generally represented in naïve and memory T cells, but not proliferating effector T cells[32,39]. SLN immunosuppression appears to transform activated CD8+ T cells into a resting-like state.

To confirm the metabolic activity modulated by aPD-1 and LDH adjuvant, we cocultured these functional agents with murine cytotoxic T lymphocytes (CTLs; activated by anti-CD3 and anti-CD28 antibodies (aCD3/28)). aPD-1, $Mg^{2+}$ or $Fe^{3+}$ (alkaline LDH cannot be used directly because it severely affects the extracellular acidification rate (ECAR) measured in this test) enabled these activated cells to exhibit long-standing glycolytic activities to diverse degrees (Fig. 3g, h). As upregulated glycolysis is intimately linked to robust T-cell immunoresponses[32], the engagement of aPD-1 and LDH would benefit the effector function of CTLs in SLNs.

## LDH adjuvant improves CTL effector function via $Mg^{2+}$/$Fe^{3+}$-mediated shaping

Unlike aPD-1, which blocks PD-1 presented on CTLs for immune regulation, the CTL-shaping mechanism by LDH adjuvant is uncertain and may be associated with the participation of $Mg^{2+}$ and $Fe^{3+}$ ions[19,20]. In the aCD3/28-activating process, either $Mg^{2+}$ or $Fe^{3+}$ enhanced the expression of CD69 (early activation marker), CD25 (activation marker), and CD107a (degranulation marker) in CTLs (Fig. 4a–c). When the MHC mismatch between BALB/c CTLs and B16F10 cancer cells was utilized for a cytotoxic test[40,41], $Mg^{2+}$ (or $Fe^{3+}$)-treated CTLs exhibited an enhanced killing capacity to targets (B16F10 cancer cells) in a 2-h killing assay (Fig. 4d). Combining both cations (i.e., LDH) enabled CTLs to kill ~20% of targets at a CTL-to-target ratio of 0.62:1. In contrast, CTLs from the control group nearly failed to work. This suggests a positive effect on CTL activation and cytotoxicity from either metal cation of the LDH adjuvant.

$Mg^{2+}$ ions have been reported to be required by the cell-surface molecule lymphocyte function-associated antigen 1 (LFA-1) for active confirmation on CD8+ T cells, boosting TCR signal transduction, metabolic programming, and even cytotoxicity[19]. LFA-1-driven signaling activity including proximal focal adhesion kinase (FAK) and distal extracellular signal-regulated kinase 1/2 (ERK1/2), could thus behave in an $Mg^{2+}$-modulated manner. As indicated in Fig. 4e, f, phosphorylated FAK (pFAK) and ERK1/2 (pERK1/2) were augmented under the stimulation of free $Mg^{2+}$ or $Mg^{2+}$-containing LDH, but not $Fe^{3+}$. Blocking the regulation with an FAK inhibitor (FAKi; PF-562271) reduced distal pERK1/2 (Fig. 4f) and ultimately CTL activation and killing performance (Fig. 4a–d). $Mg^{2+}$ cations from the LDH adjuvant shape CTL effector function via the $Mg^{2+}$-pERK1/2 axis (Fig. 4i).

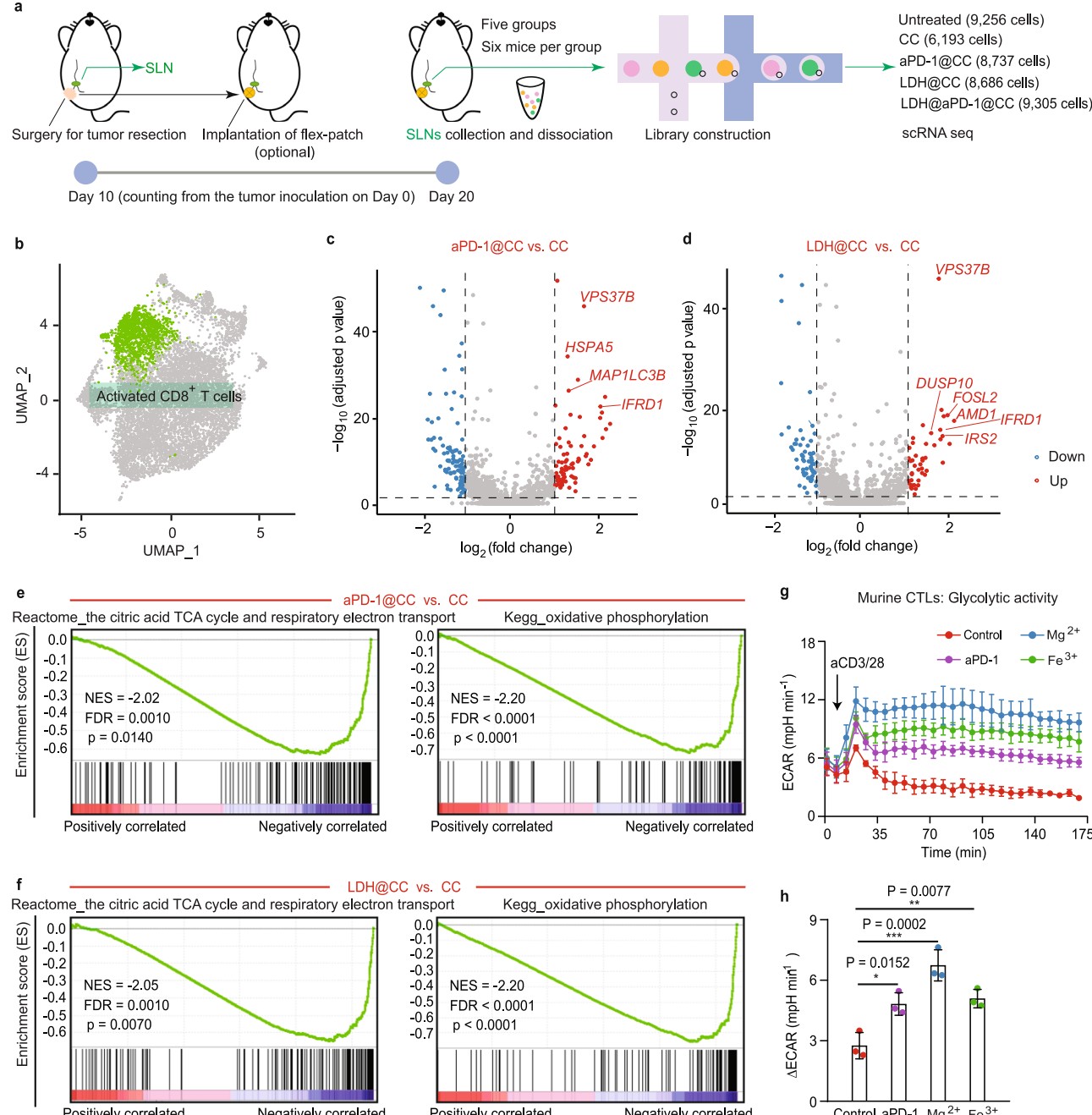

**Fig. 3 | Regulated metabolic activities of activated CD8$^+$ T cells in SLNs after aPD-1 and LDH treatment. a** Schematic illustrating the scRNA seq trial on the 4T1 BC mouse model after surgery, including no patch implantation (untreated) and patches with different therapeutic agents (CC, aPD-1@CC, LDH@CC, and LDH@aPD-1@CC). $n = 6$ mice per group. **b** UMAP plot highlighting the identified activated CD8$^+$ T cells. Volcano plots showing DEGs in activated CD8$^+$ T cells between aPD-1@CC vs. CC (**c**) and LDH@CC vs. CC (**d**). Significant DEGs (blue and red dots): $p$ value adjusted <0.05 and fold change >2. Representative GSEA-enriched pathways for activated CD8$^+$ T cells in aPD-1@CC vs. CC (**e**) and LDH@CC vs. CC (**f**). **g, h** Glycolytic activity of murine CTLs with diverse treatments. Glycolytic activity (ΔECAR) was determined by subtracting the ECAR maximum from baseline. $n = 3$ per group. Statistical significance was calculated via one-way ANOVA with Tukey's multiple comparisons. *$p < 0.05$, **$p < 0.01$, and ***$p < 0.001$. For (**g, h**), data are presented as the mean ± SD. Source data are provided as a Source Data file.

Moreover, cellular ROS participate in the modulation of TCR signaling of CTLs via a nuclear factor of activated T cell (NFAT)-related pathway[20,21]. Stimulating CTLs with ROS-promoter $Fe^{3+}$ or $Fe^{3+}$-involving LDH, we identified the translocation of NFAT1 from the cytoplasm to the nucleus (Fig. 4g), suggesting that NFAT was activated and acted as a transcription factor. Nuclear import of NFAT1 was not obviously found in the $Mg^{2+}$-treated group but was downregulated upon addition of a ROS consumer (glutathione, GSH), which confirms the ROS-relevant function of $Fe^{3+}$ in NFAT activation. Inhibiting activation with an NFAT inhibitor (NFATi; 11R-VIVIT TFA) interfered with downstream

IL-2 induction (Fig. 4h) and the contribution of LDH to CTL effector function (Fig. 4a–d). $Fe^{3+}$ cations from the LDH adjuvant program CTL immunity through the ROS-NFAT axis (Fig. 4i). Taken together, LDH adjuvant promotes CTL effector functions involving activation, glycolytic activity, and cytotoxic killing via metal cation-mediated shaping.

**Flex-patch shapes SLN antitumor immunity**
Given the foregoing functional mechanism, we harvested SLNs where flex-patch works to assess the immunostimulatory effects in vivo. Rapid

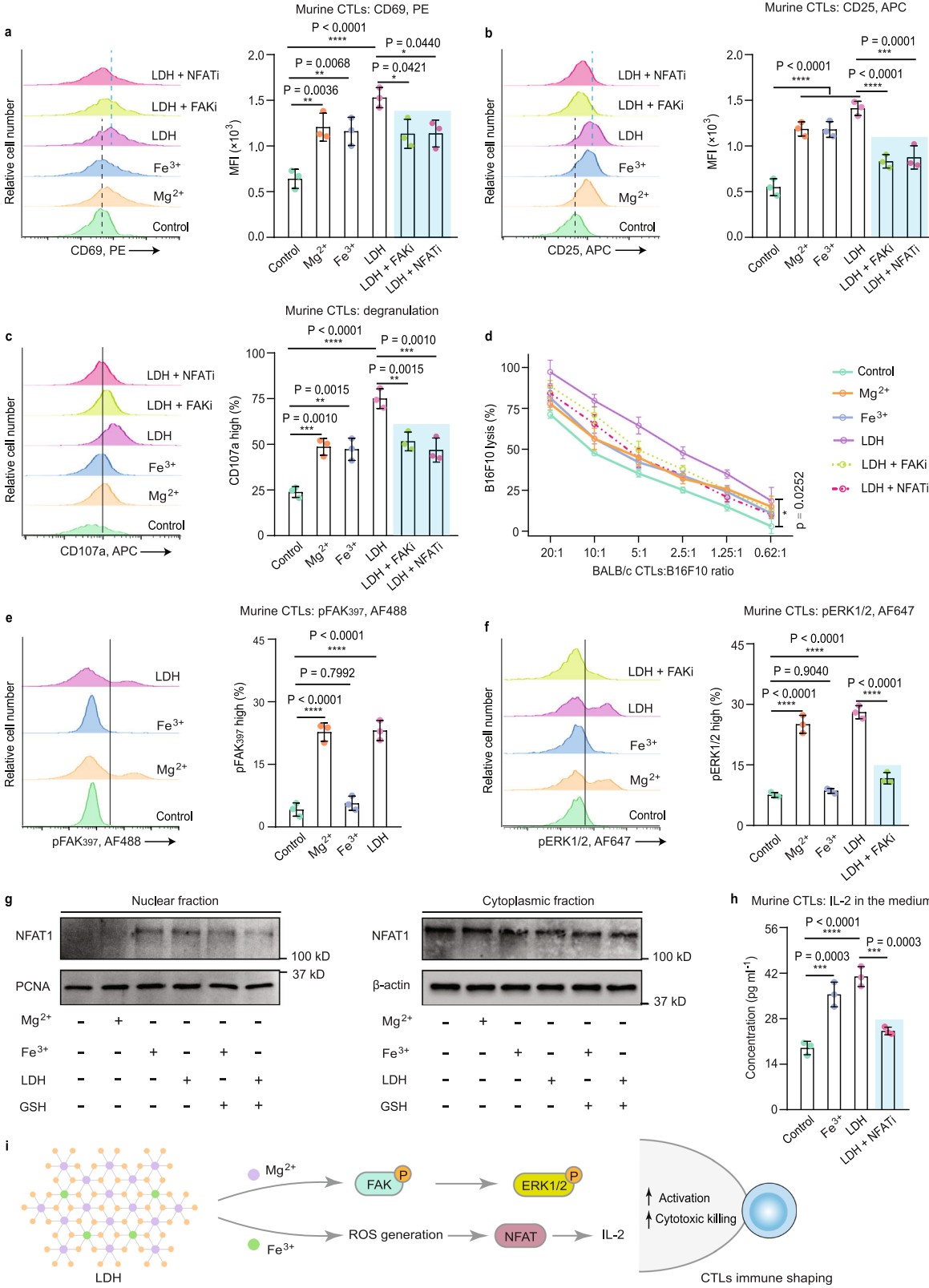

**Fig. 4 | In vitro activation and cytotoxicity of CTLs regulated by LDH adjuvant.** Representative histograms and quantification of the expression of the early activation marker CD69 (**a**), activation marker CD25 (**b**), and degranulation marker CD107a (**c**) on murine CTLs. **d** A 2-h killing assay of CTLs with varying ratios of BALB/c CTLs to targets (B16F10 cancer cells). Representative histograms and quantifications of the phosphorylation of FAK (**e**) and the phosphorylation of ERK1/2 (**f**); **g** NFAT1 (also known as NFATc2, nuclear factor of activated T cells,

cytoplasmic 2) levels in nuclear and cytoplasmic fractions, investigated by Western immunoblotting. **h** CTL IL-2 levels in the cellular media. **i** Representative pathways involved in CTL immune shaping by the LDH adjuvant. For all tests, $n = 3$ per group. For (**a**–**f**, **h**), data are presented as the mean ± SD, and statistical significance was calculated via one-way ANOVA with Tukey's multiple comparisons, *$p < 0.05$, **$p < 0.01$, ***$p < 0.001$, and ****$p < 0.0001$. Source data are provided as a Source Data file.

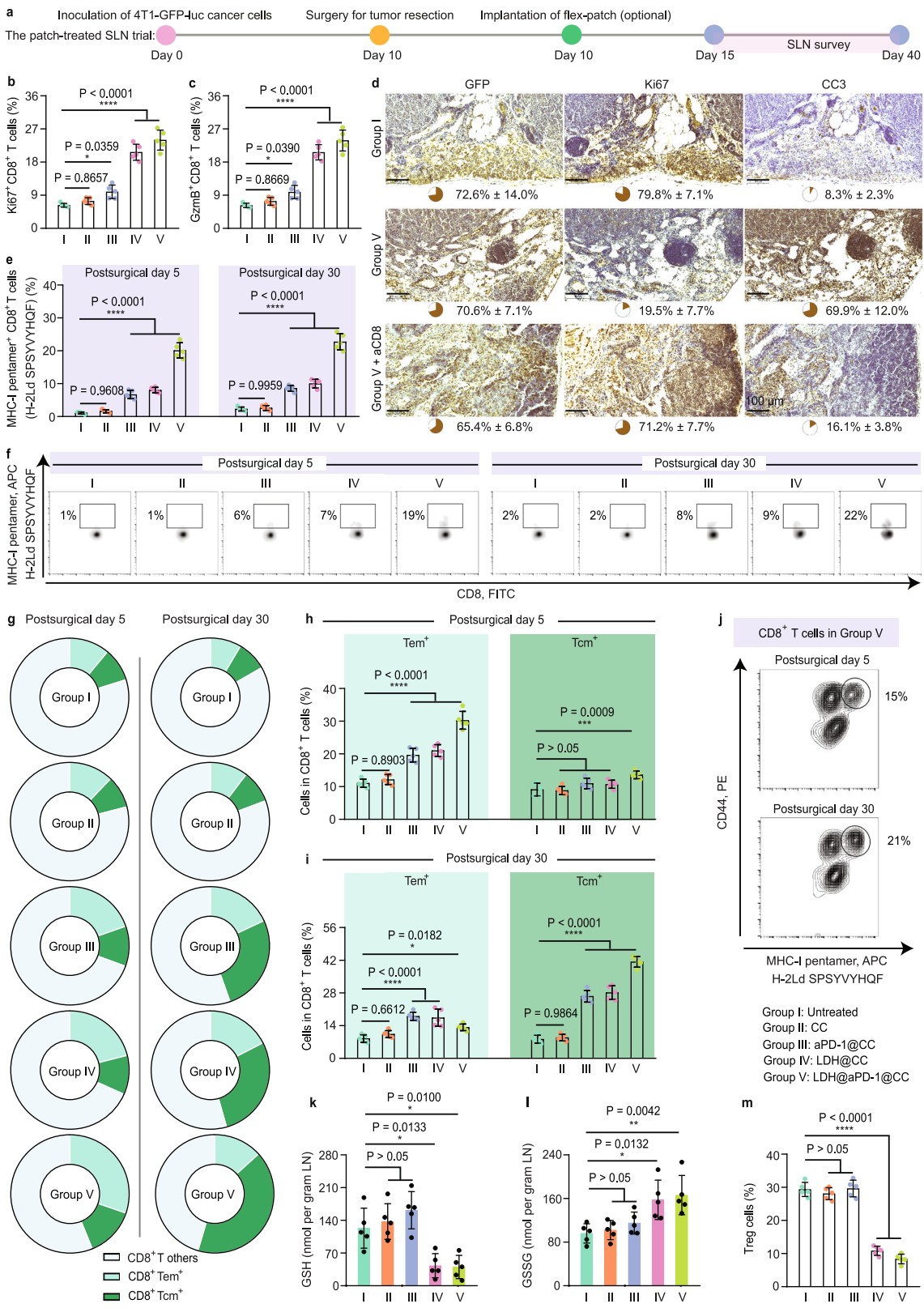

increases in the proliferation (Ki67[+]) and killing effect (indicated by the expression of cytotoxic granzyme B (GzmB[+]))[41] of CTLs were quantified in groups III-V (aPD-1@CC, LDH@CC, and LDH@aPD-1@CC) on day 5 postsurgery (Fig. 5a−c and Supplementary Fig. 11 (gating strategy), and Supplementary Fig. 12a, b). Among these groups, the LDH adjuvant amplified the functional effect by approximately twofold more than aPD-1. Combining both promoters of CD8[+] T activities triggered apparent

apoptosis (indicated by high levels of cleaved caspase-3 (CC3), an apoptosis marker) of invasive 4T1 cancer cells (labeled with GFP) in SLN (Group V, Fig. 5d). When CD8 antibodies (aCD8) were intraperitoneally injected into patch-treated mice, high proliferation (strong Ki67 level) of tumor cells was recovered (Group V + aCD8, Fig. 5d). Thus, flex-patch treatment induced tumoral apoptosis in metastatic SLNs, which was associated with the enhanced effector function of CD8[+] T cells.

**Fig. 5 | Antitumor immune shaping of flex-patch to immunosuppressive SLNs.**
SLNs in each group were harvested on postsurgical days 5 and 30. **a** Schematic illustrating the SLN survey on the 4T1 BC mouse model after surgery, including no patch implantation (untreated, Group I) and patches with different therapeutic agents (CC, Group II; aPD-1@CC, Group III; LDH@CC, Group IV; and LDH@aPD-1@CC, Group V). Flow cytometric quantification of Ki67[+]CD8[+] T cells (**b**), GzmB[+]CD8[+] T cells (**c**), H-2Ld SPSYVYHQF-pentamer[+]CD8[+] T cells (**e**) and their representative flow cytometric plots (**f**), gating on CD8[+]CD3[+] cells in SLN.
**d** Immunohistochemistry (IHC) staining and quantification of GFP (4T1 marker), Ki67 (proliferation marker), and CC3 (apoptosis marker) in SLN. Scale bar: 100 μm;
$n = 3$ mice per group. Proportional variation (**g**) and flow cytometric quantification (**h**, **i**) of effector memory ($T_{em}$; CD62L⁻CD44⁺) and central memory ($T_{cm}$; CD62L⁺CD44⁺) CD8⁺ T cells, gating on CD8[+]CD3[+] cells in SLN. **j** Representative flow cytometric plots of CD44[+]H-2Ld SPSYVYHQF-pentamer[+]CD8[+] T cells, gating on CD8[+]CD3[+] cells in SLN. GSH concentration (**k**) and GSSG concentration (**l**) of SLN tissue on day 5 postsurgery. **m** Flow cytometric quantification of Foxp3[+]CD25[+]CD4[+] T cells, gating on CD4[+]CD3[+] cells in SLN on day 5 postsurgery. For (**b**, **c**, **e**–**m**), $n = 5$ mice per group; statistical significance was calculated via one-way ANOVA with Tukey's multiple comparisons, *$p < 0.05$, **$p < 0.01$, ***$p < 0.001$, and ****$p < 0.0001$; data are presented as the mean ± SD. Source data are provided as a Source Data file.

Next, to identify the tumor-specific immunomodulation that flex-patch brought in, an AH1 pentamer was custom made. AH1, an H-2Ld-restricted peptide with the sequence SPSYVYHQF, is an immunodominant antigen shared by several murine tumors, including 4T1[42–44]. Of the treated SLNs, H-2Ld SPSYVYHQF pentamer[+]CD8[+] T cells were distinguished and exhibited rapid and robust enhancement from postsurgical day 5 to day 30, especially in the LDH@aPD-1@CC-treated SLNs (Fig. 5e, f; pentamer loaded with H-2Ld MVPGGQSSF peptide, an immunodominant epitope originating from *Mycobacterium tuberculosis*[45], was customized as a control (Supplementary Fig. 12c)). As a comparison, no H-2Ld SPSYVYHQF pentamer[+]CD8[+] T cells were traced in the contralateral LNs (Supplementary Fig. 13). Both tests emphasized the importance of metastatic SLN and the high efficacy of our LDH@aPD-1@CC flex-patch on tumor-specific CD8⁺ T activation.

While regulating SLN, immunological memory is the foundation of acquired antitumor immunity[46]. In the patch-driven month, differentiation of memory CD8⁺ T cells was identified (Fig. 5g–i and Supplementary Fig. 12d). On day 5 postsurgery, effector memory CD8⁺ T cells ($T_{em}^+$, CD62L⁻CD44⁺) in Group V exhibited over twofold mounting compared with those in Group I. On day 30 postsurgery, although there were decreased CD8⁺ $T_{em}^+$ cells (Group V), central memory CD8⁺ T cells ($T_{cm}^+$, CD62L⁺CD44⁺) increased approximately fivefold compared with the untreated cells. According to previous studies, $T_{em}$ cells display immediate effector function in response to antitumor therapy, whereas $T_{cm}$ cells mediate reactive memory, proliferation and differentiation into effector T cells if antigen stimulation occurs again[47,48]. The slow-release feature of the LDH@aPD-1@CC patch could long-termly maintain a high proportion of memory CD8⁺ T cells (both $T_{em}^+$ and $T_{cm}^+$, Fig. 5g) in the SLN, especially CD8⁺ T cells with antigen-specific immune memory (CD44[+]H-2Ld SPSYVYHQF pentamer[+], Fig. 5j), fighting against cancer resurgence.

Additionally, during the engagement of SLNs with LDH-containing patches (groups IV and V, Fig. 5k, l), we found intra-SLN GSH oxidation into GSSG (oxidized glutathione). This redox modulation covered SLN CD8⁺ and CD4⁺ T cells (Supplementary Fig. 14a, b). For CD8⁺ T cells, an activating and cytotoxic enhancement driven by LDH-mediated ROS was demonstrated (Fig. 4). Of CD4⁺ T cells, this inorganic adjuvant decreased the Treg (Foxp3[+]CD25[+]CD4[+]) proportion by one-third of the single aPD-1 treatment (Group IV vs. Group III; Fig. 5m and Supplementary Fig. 14c). According to a recent Treg discovery, high serine metabolism downregulates Treg Foxp3 expression. This serine metabolism, however, is generally restricted by GSH to maintain Treg functionality[49]. Unfreezing the restriction/protection from GSH could impact Treg Foxp3. Consistent with this report, consuming GSH via LDH adjuvant in our case reduced Foxp3 levels in Tregs, which was also prominent in the scRNA seq trial (Supplementary Fig. 15).

### Flex-patch performs efficient postsurgical immunoadjuvant therapy

To investigate the postsurgical immunoadjuvant performance of flex-patch in vivo, patches loaded with different functional agents were implanted after primary tumor dissection on day 10 of 4T1-GFP-luc inoculation (Fig. 6a). In vivo and ex vivo bioluminescence (Fig. 6b and Supplementary Fig. 16) and tumor volume monitoring (Fig. 6c) showed that mice with surgery only (Group I) or with the CC matrix (Group II) had severe tumor relapse. The aPD-1@CC (Group III) and LDH@CC (Group IV) patches protected approximately half of the 12 mice from tumor recurrence for at least 1 month; nevertheless they were not comparable with the effect of LDH@aPD-1@CC (Group V), in which 10 out of 12 mice were tumor-free (Fig. 6b–e). Examination of postsurgical tumors showed that treatment with the LDH@aPD-1@CC patch allowed most CTL infiltration in the postsurgical debris (Fig. 6f–h and Supplementary Fig. 17). These CTLs were responsible for the elimination of remaining 4T1 cancer cells (Fig. 6i), which otherwise remained alive if CTLs were blocked with aCD8. Regulating CTLs in SLN by the implanted LDH@aPD-1@CC patch impeded postsurgical 4T1 tumor relapse highly efficiently.

During the treatment, the body weights of all mice were not obviously influenced (Supplementary Fig. 18). Moreover, although much higher levels of pro-inflammatory TNF-α, IL-12, and interferon γ (IFN-γ) cytokines and decreased anti-inflammatory IL-10 in sera were observed on day 5 postoperation (Supplementary Fig. 19a, b), all were rapidly remodeled during the following 10 days (Supplementary Fig. 19c). This may explain the unaffected wound healing in groups II-V compared with Group I (Supplementary Fig. 19d).

To next test the long-term antitumor protection that flex-patch evoked, we rechallenged tumor-free mice collected from Group V with 4T1-GFP-luc cells (Fig. 7a). During the following 30 days, distinguishing from the zero survival in the control group, all mice after LDH@aPD-1@CC therapy successfully resisted tumor development (Fig. 7b–d). However, intraperitoneal injection of aCD8 completely subverted the tumor elimination efficacy.

To investigate a broad-spectrum application of flex-patch, we further established an EMT6 BC model (Supplementary Fig. 20a). Surgery and patch implantation were adopted when the primary tumor grew to ~500 mm³ on day 12 postinoculation. When the therapeutic effect in each group was surveyed, the LDH@aPD-1@CC patch likewise exhibited the best adjuvant performance and inhibited all tumor recurrence (Supplementary Fig. 20b–e). In the following rechallenge trial, 87.5% of LDH@aPD-1@CC-treated mice remained tumor free, which unsurprisingly was invalidated by aCD8 blocking (Supplementary Fig. 20f–i). Together, aPD-1, in concert with the LDH immune adjuvant, aroused in the postsurgical host an efficient and long-term therapeutic defense against BC relapse, and this defense was associated with patch-reshaped CTLs.

## Discussion

To date, resecting metastatic SLNs remains an intractable event in clinical tumor surgery. This explains these ongoing clinical trials tracking survival in patients assigned to either SLN dissection or no SLN dissection[3–5]. However, for the no-dissection group, metastatic SLN was simply underestimated as a postoperative residual. This small bean-shaped tissue may work as an immunoadjuvant arena from our perspective, consistent with recent studies on the antitumor significance of tumor-draining lymph nodes[8,16]. In this context, unlike general post-adjuvant strategies eliminating SLN-tumor cells via chemical or radiation-initiated approaches that cannot benefit patients with immunosurveillance, we propose an immunotherapeutic

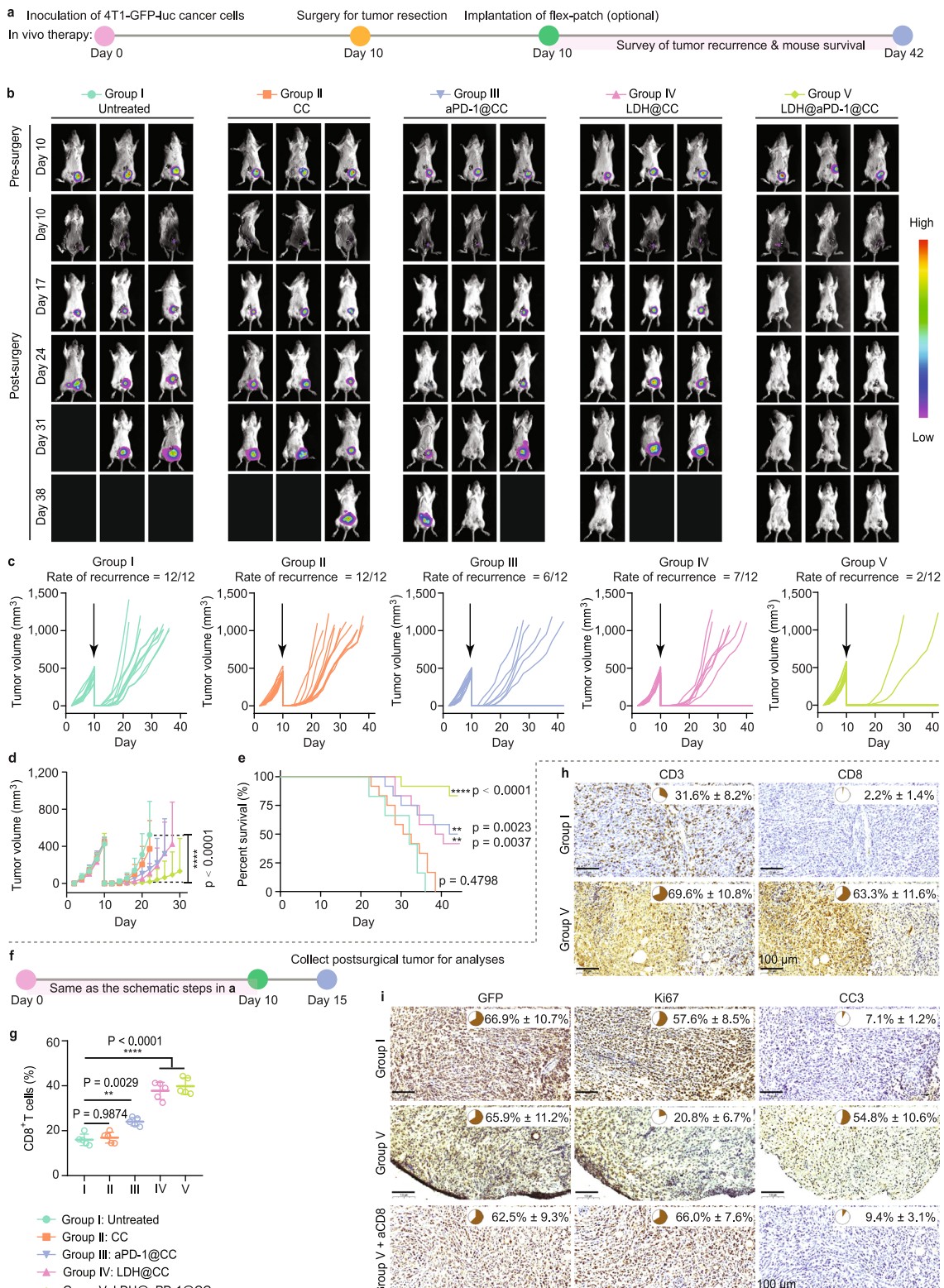

**Fig. 6 | Postsurgical 4T1 BC treatment of flex-patch. a** Schematic illustrating therapeutic procedure in the 4T1 BC mouse model after surgery, including no patch implantation (untreated, Group I) and patches with different therapeutic agents (CC, Group II; aPD-1@CC, Group III; LDH@CC, Group IV; and LDH@aPD-1@CC, Group V). $n = 12$ mice per group. **b** Representative bioluminescence images of the mice per group. Individual (**c**) and average (**d**) tumor growth kinetics per group. **e** Mouse survival per group. For (**b**–**e**), $n = 12$ mice per group. **f** Schematic illustrating the analyses of postsurgical 4T1 tumors. $n = 8$ mice per group. **g** Flow cytometric quantification of CD8$^+$ T cells, gating on CD3$^+$CD45$^+$ cells ($n = 5$ mice per group). IHC detection and quantification of infiltrated CD3$^+$ T and CD8$^+$ T cells (**h**) and 4T1 tumoral proliferation and apoptosis (**i**; GFP, 4T1 marker; Ki67, cell proliferation marker; and CC3, cell apoptosis marker) in postsurgical tumor tissue. Scale bar: 100 μm; $n = 3$ mice per group. For (**d, g**), statistical significance was calculated via one-way ANOVA with Tukey's multiple comparisons. For (**e**), statistical significance was calculated via the log-rank (Mantel–Cox) test by comparison with the untreated Group I. $^{**}p < 0.01$ and $^{****}p < 0.0001$. For (**d, g**–**i**), data are presented as the mean ± SD. Source data are provided as a Source Data file.

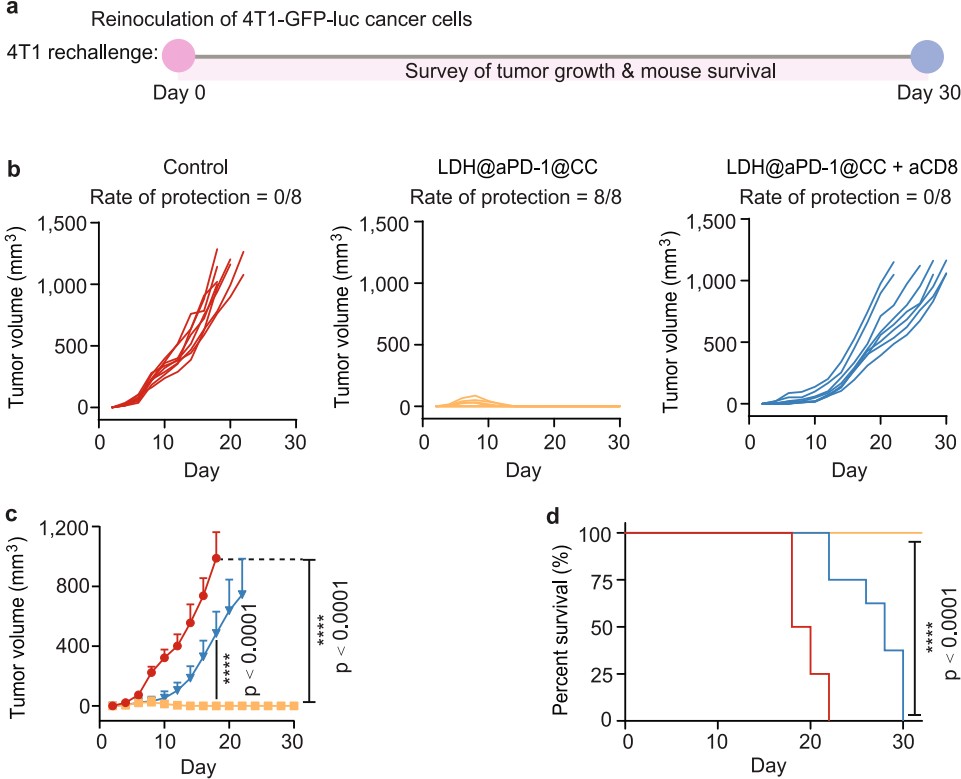

**Fig. 7 | 4T1-tumor rechallenging for patch-treated mice. a** Schematic illustrating the route of 4T1 tumor rechallenge assay. Apart from the control group, all tumor-free mice were harvested from Group V (LDH@aPD-1@CC). $n = 8$ mice per group. Individual (**b**) and average (**c**) tumor growth kinetics per group. **d** Survival of mice in different rechallenge groups. For (**c**), the curve ended when the first mouse in the corresponding group died; statistical significance was calculated via one-way ANOVA with Tukey's multiple comparisons; data are presented as the mean ± SD. For (**d**), statistical significance was calculated via the log-rank (Mantel–Cox) test by comparison with the control group. ****$p < 0.0001$. Source data are provided as a Source Data file.

flex-patch for post-surgery care. The flex-patch launched immunoactive agents into metastatic SLN spatiotemporally and reshaped CTLs to fight against peripheral cancer residuals. This immune clearance process served to initiate in situ tumor vaccination in metastatic SLN, conferring patients with robust and long-term tumor-specific immune protection (Figs. 5–7). Moreover, loaded agents of flex-patch could be screened and combined flexibly in a wide range of clinical approval, which allows the patch to be compatible with university and customization based on cancer complexity.

Previous studies have revealed the antigen-presenting dysfunction of SLN CTLs because of tumor-driven immunosuppression. Here, to study them via scRNA seq and a glycolytic metabolism kit, we identified a correlation between immune dysfunction and aberrant mitochondrial activity. These activated CD8+ T cells enriched the TCA cycle and OXPHOS pathways, which are common in the mitochondria of resting/naïve cells and cannot support the bioenergetic requirements for killing. When regulating metastatic SLN with the flex-patch, the aforementioned phenomena were unrecognized but replaced with enhanced glycolytic activity. Naïve CD8+ T cells face metabolic changes upon activation. However, here, we understood that the metabolic state of activated CD8+ T cells was not constant, but rather dynamically varied during immune programming. These variations may correlate with cellular mitochondrial activity.

Furthermore, the irreplaceable significance of immune adjuvant LDH is emphasized in SLN-aided antitumor immunity. With a licensed aluminum hydroxide (immune adjuvant for use in humans)-like molecular structure, LDH was further modified to maintain a nanosheet morphology and two immune-functional cations (magnesium and iron). These characteristics guarantee safe and efficacious SLN accumulation and in situ immune regulation by the nanoadjuvant.

Regarding the adjuvant's contribution to CTLs, it boosted the activation and cytotoxic killing of these CTLs through $Mg^{2+}$- or $Fe^{3+}$-mediated strategies. This direct immune regulation may inform LDH nanoadjuvant high potential in cell-mediated immunity and an extended application in tumor vaccination area.

In summary, an easily engaged flex-patch was shown to leverage a unique postoperative window, rescuing multiple postsurgical issues concurrently. It presents a solution to the metastatic SLN-resecting dilemma in clinical surgery, reshapes SLN remnants for immunoadjuvant therapy, and protects BC models from high-incidence tumor recurrence in a robust and sustained mode. This flex-patch established a positively correlated bridge between SLN immunoreshaping and postsurgical immunotherapy.

## Methods

### Breast tumor model with orthotopic metastasis

All mouse studies were carried out following the protocols (protocol ID: UMARE-030-2021 and UMAEC-015-2019) approved by the Animal Ethics Committee in University of Macau (UM). All mice were maintained under specific pathogen free (SPF) conditions in the UM animal facility. This study was not sex/gender-specific. Although breast cancer is generally diagnosed among females, males can develop it.

To establish breast tumor model with orthotopic metastasis, mouse metastatic 4T1 cancer cells (2 million) were surgically inoculated into the 4th mammary gland (right-sided) of syngeneic host (BALB/c, female, 8-week age), who was anesthetized with avertin (125 mg per kg body weight). Wound clips were used for skin suture. Heating pad was used during the whole process. Tumor was monitored and measured with digital caliper every other day. Tumor volume was calculated using the formula: volume $= \frac{4}{3}\pi \times \left(\frac{x}{2}\right)^2 \times \left(\frac{y}{2}\right)$, where $x$, the

longest dimension of the tumor and $y$, the dimension perpendicular to $x$. When measuring, mice were immediately euthanized if their breast tumor volume reached up to 1000 mm³. Here, maximal tumor burden permitted by the UM Animal Ethics Committee (protocol ID: UMARE-030-2021) is 1300 mm³. EMT6 breast tumor model was established and applied with the same approach.

## Reagents

Reagents used in this study are listed in Supplementary Tables 1–4 with all corresponding details.

## Fabrication and characterization of flex-patch

Flex-patch had carboxylated chitosan (CC) as the patch framework, entrapping anti-mouse PD-1 antibodies (aPD-1) and layered double hydroxide (LDH) nanoadjuvants. LDH we synthesized is magnesium iron(III)-based LDH with a molar ratio of four to one. In detail, aquatic solution (5 ml) dissolving magnesium nitrate hexahydrate (4 mmol) and iron(III) nitrate nonahydrate (1 mmol) was added into 20 ml of alkaline solution containing sodium hydroxide (10 mmol) rapidly. After half an hour of vigorous stirring at room temperature condition, fresh LDH slurry was collected by centrifugation ($2400 \times g$ for 5 min) and twice washing with 20 ml of deionized water. Re-dispersing the slurry into 20 ml of deionized water generated ~15 mg ml⁻¹ of LDH suspension (equal to 2 mg ml⁻¹ of Fe) for experimental use. The sheet-like morphology of LDH was imaged by transmission electron microscopy (TEM; Jeol JEM-ARM300F). Size distribution and zeta potential of these nanoparticles were measured via a Zetasizer 8.01 (Malvern Panalytical).

Commercial CC shows a negative charge (–34.0 ± 6.8 mV) due to enriched carboxylate groups, integrating aPD-1 and positive-charge LDH nanoparticles via electrostatic interaction easily. In our tests, 10 ml of 60 mg ml⁻¹ CC aquatic solution was prepared firstly by dissolving CC in deionized water. CC solution was dropped into the lid well of 48-well plate (100 ul per well), drying in oven (35 °C) for 1 h, to get rounded CC patch with 10-mm diameter successfully. Following the same procedure, ultrasonic mixing of aPD-1 (50 μg) and CC solution (100 ul) thoroughly obtained aPD-1@CC patch; integrating LDH suspension (200 ul) and CC solution (100 ul) achieved LDH@CC patch; packaging aPD-1 (50 μg), LDH suspension (200 ul), and CC solution (100 ul) acquired LDH@aPD-1@CC patch. Distribution of loaded aPD-1 was fluorescently imaged by Leica DFC450 C, with PE fluorophore labeling antibodies. Microscopic morphology of the LDH@aPD-1@CC patch was examined by scanning electron microscopy (SEM, Hitachi S-3400N, Type I) and energy dispersive X-ray spectroscopy (EDS, Horiba EX-250).

## Tests about aPD-1 (or LDH) releasing from patch

To test aPD-1 releasing from patch, LDH@aPD-1@CC patch was immersed into 2 ml of pH 6.5 phosphate buffer (containing 0.0357 M $Na_2HPO_4$ and 0.0643 M $NaH_2PO_4$) to simulate the acidic postoperative wound. Working buffers were collected time-dependently by centrifugation, to quantify the released aPD-1 via a mouse IgG ELISA kit. Same phosphate buffer was applied to detect LDH releasing from LDH@aPD-1@CC patch and LDH skeleton decomposition. Released nanoparticles were washed, lyophilized, and then collected for X-ray powder diffraction (XRD) and Fourier transform infrared (FTIR, IRAffinity-1S, SHIMADZU) spectroscopy analyses, to demonstrate the material composition. Disintegration of LDH skeleton was tested by using a Slide-A-Lyzer MINI dialysis device (3.5 K MWCO). As the manufacturer's procedure suggested, 0.5 ml of LDH suspension was kept in conical tube and was treated by 14 ml of pH 6.5 phosphate buffer. Membrane with 3.5 K MWCO guarantees free ions ($Mg^{2+}$ and $Fe^{3+}$) disintegrated from LDH framework to penetrate only. Quantitative $Mg^{2+}$ and $Fe^{3+}$ ions were determined by inductively coupled plasma mass spectrometry (ICP-MS).

## Labeling LDH with Cy5.5

To image patch in vivo, LDH nanoadjuvants were labeled by fluorophore Cyanine-5.5 (Cy5.5) in a two-step method. Fluorophore Cy5.5 was firstly conjugated with a peptide (SPSYVYHQF, hydrophilic peptide with isoelectric point of pH 7.66), known as peptide-Cy5.5. The peptide-Cy5.5 has high affinity to positive-charge LDH, guaranteeing the labeling stability. Briefly, 1 mg of peptide-Cy5.5 was dissolved into 100 μl of DMSO, and was further mixed into 4 ml of nitrate-containing LDH suspension (7.5 mg ml⁻¹, pH -10) to be anionic. Anionic peptide-Cy5.5 were easily exchanged with interlayer nitrate and loaded into LDH, forming LDH/peptide-Cy5.5 nanoparticles, abbreviated as LDH-Cy5.5. After 1-h gentle stirring, twice washing was continued to collect LDH-Cy5.5 slurry via centrifugation ($9400 \times g$ for 10 min). Re-dispersing LDH-Cy5.5 slurry into 2 ml of deionized water generated ~15 mg ml⁻¹ of LDH-Cy5.5 suspension for experimental use. Successful loading was confirmed with XRD. Loading efficacy was ~80%, determined by Pierce BCA protein assay kit. Same patch preparation approach was adopted to assemble aPD-1 (50 μg), LDH-Cy5.5 (200 ul) suspension, and CC solution (100 ul) to acquire LDH-Cy5.5@aPD-1@CC flex-patch, for intravital imaging of patch dissembling and body destination in mouse model after breast cancer surgery.

## Cell culture

4T1 and EMT6 breast cancer cells were transfected with green fluorescent protein (GFP) and luciferase, labeled as 4T1-GFP-luc and EMT6-GFP-luc, respectively. The 4T1, 4T1-GFP-luc, EMT6, or EMT6-GFP-luc cancer cells were cultured in Roswell Park Memorial Institute (RPMI) 1640 medium supplemented with 10% (v/v) of fetal bovine serum (FBS) and 1% (v/v) of penicillin-streptomycin at 37 °C in an incubator with 5% (v/v) of carbon dioxide. Same culturing parameters were applied to RAW264.7 cell line, except the cultural medium of Dulbecco's Modified Eagle's Medium (DMEM). All cell lines were tested for mycoplasma free.

## In vivo implantation, disassembling and biodistribution of flex-patch

To study the in vivo implantation, disintegration and body destination of flex-patch, breast tumor model was firstly established by inoculating 4T1 cells (2 million) orthotopically, with the approach described above. Briefly, mice were anesthetized with avertin (125 mg per kg body weight). Sterilized surgical tools were used to remove solid tumor on post-inoculation day 10. Flex-patch was then attached on wound bed immediately. The implanting process was imaged by Leica DFC450 C microscopy. Wound clips were used for skin suture.

To test in vivo disassembling of flex-patch, mice with postsurgical implantation of LDH-Cy5.5@aPD-1@CC were imaged using an IVIS Spectrum Imaging system (PerkinElmer) periodically (excitation: 680 nm; emission: 700 nm). Region of interest was quantified through the PerkinElmer's portfolio of imaging software.

To analyze body destinations of flex-patch, main tissues including SLN and its opposite LN, heart, spleen, liver, lung, kidney, muscle, brain, blood, and gut were collected periodically after patch implantation. Iron concentrations in those tissues were measured by ICP-MS, after an acid digestion of fresh nitric acid and hydrogen peroxide mixture (4:1, v/v). Given that the original irons in animal tissues, iron concentrations before patch implantation were set as the baseline of "zero".

## In vitro and in vivo biosafety of the patch

Two cell lines (RAW264.7 and 4T1 cells) were used to test the cytotoxicity of functional components from patch. Different concentrations of CC (or LDH nanoparticles) were co-cultured with 0.1 million of RAW264.7 (or 4T1) cells in 48-well plate for 48 h, separately. MTT assay was followed then.

In vivo biosafety of patch was evaluated in postsurgical 4T1 tumor model introduced in Methods. On day 36 after LDH@aPD-1@CC patch implantation, mouse organs including live, kidney, spleen, heart, and lung were collected, to characterize their histopathological difference to healthy mice using a hematoxylin and eosin (H&E) staining. At the same time, blood sera from healthy mice and patch-treated mice were harvested, to quantify the levels of six key biochemical markers (BUN, blood urea nitrogen; CK, creatine kinase; LADH, lactate dehydrogenase; CRE, creatinine; AST, aspartate transaminase; ALT, alanine aminotransferase) through the corresponding assay kits.

## Identifying tumor metastasis in SLN
To identify SLN metastasis, same orthotopic breast tumor model was established by inoculating 4T1-GFP-luc cells (2 million). On post-inoculation day 10, representative LNs (two brachial LNs, two axillary LNs, and two inguinal LNs) were harvested for an immediate bioluminescence imaging and GFP-associated immunohistochemistry (IHC) staining by following the manufacturer's instructions.

## Postsurgical BC treatment of flex-patch
To explore the therapeutic performance of flex-patch to curb BC recurrence after surgery, breast tumor model was established by inoculating 4T1-GFP-luc cells (2 million) orthotopically, with the approach aforementioned. On day 10 post inoculation, mice were divided into five groups randomly ($n = 12$ per group), with tumor volume reached 500 cm³ on average. On the same day, primary tumor surgery was carried out. Briefly, mice were anesthetized with avertin (125 mg per kg body weight). Sterilized surgical tools were used to remove solid tumor, leaving ~1% (w/w) tumor to mimic postsurgical residual. Flex-patch was attached on postsurgical wound: group I, untreated; group II, CC only; group III, aPD-1@CC; group IV, LDH@CC, group V, LDH@aPD-1@CC, with calculated dosages of CC (300 mg per kg body weight), aPD-1 (2.5 mg per kg body weight), and LDH (150 mg per kg body weight). Wound clips were used for skin suture. Heating pad was used during the whole process.

During postsurgical 32 days, body weight and tumor volume of mice in each group were monitored every other day. Tumor relapse was assessed by bioluminescence imaging once per week. To collect all mice without recurrence in group V, we challenged these survivals with 4T1-GFP-luc cells inoculation. 4T1-GFP-luc cells (1 million) were surgically inoculated into the 4th mammary gland (left-sided) and were divided into two groups ($n = 8$ per group): one without treatment; another one with peritoneal anti-CD8 antibodies (aCD8) injection (twice with 200 μg per injection; post-inoculation day 1 and 3). Tumor volume of mice in each group were monitored every other day. In 4T1 challenging trial, a new group of BALB/c female mice ($n = 8$) was added to get the tumor cells inoculation, as a control.

To study the broad application of our flex-patch, another breast tumor model was established by inoculating EMT6-GFP-luc cells (2 million) orthotopically, with the same strategy.

## Single-cell RNA sequencing and data processing
We ran single-cell RNA sequencing (scRNA seq) on SLN single-cell suspension on postsurgical day 10 ($n = 6$ per group), using the Chromium Next GEM Single Cell 3′ Reagent Kits v3.1 from 10x Genomics based on the manufacturer's instructions. Single-cell transcriptome libraries were sequenced on an Illumina HiSeq X-Ten system, and were mapped to expended mm10 mouse reference genome using CellRanger tookit (v4.0.0, 10x Genomics). Quality control was processed to all samples. All cells expressing <200 or >6000 genes were removed; cells with high-level mitochondrial gene expression were also filtered out through estimating a median-absolute deviation (MAD) variance with median centered; the DoubletFinder (2.0.3) with default parameters were used to identify and remove the potential doublets. Finally, total 39,155 cells passed quality control.

The above pre-processed data were next submitted to Seurat (v3.9), proceeding unique molecular identifier (UMI) count matrix normalization with default parameters. Functions of NormalizaData and FindVariableFeatures were applied to identify variable features, respectively. IntegrateData function integrated all samples. ScaleData enabled each gene an equal weight, to avoid that genes with high expression dominate the downstream analysis. The linear dimensional reduction and graphing-based clustering were used to group cells. We defined and annotated each cell cluster via SingleR (v1.4.0). Also, FindMarkers and database searching confirmed the marker genes per cluster.

We then applied R package DESeq2(v.1.30.0) to analyze different expression genes (DEGs), and used gene set enrichment analysis (GSEA) to identify metabolic pathways enriched in genes with highest variability. GSEA analysis was done by the software javaGSEA available at https://www.gsea-msigdb.org/gsea/index.jsp.

## In vitro assays of murine CTLs
Naïve CD8+ T cells were isolated from the single cell suspension of BALB/c spleen via a magnetic bead-based isolation kit. The corresponding CTLs were activated with soluble anti-CD3 antibodies (5 μg ml⁻¹) and anti-CD28 antibodies (2.5 μg ml⁻¹) by culturing these naïve cells in the RPMI 1640 medium supplemented with 10% (v/v) of heat-inactivated FBS, 1% (v/v) of penicillin-streptomycin, 1 mM sodium pyruvate and 50 μM β-mercaptoethanol. To test the expression of the early activation marker (CD69), activation marker (CD25), and degranulation marker (CD107a), MgCl₂ (1 mM), FeCl₃ (0.05 mM), LDH (Mg²⁺, 1 mM), LDH (Mg²⁺, 1 mM) + FAK inhibitor (FAKi, 4 nM), or LDH (Mg²⁺, 1 mM) + NFAT inhibitor (NFATi, 50 μM) was replenished in the CTL (0.5 million per well) activating process for 8 h. To analyze the FAK and ERK1/2 phosphorylation, same activating additives and culturing time were adopted prior to the fixation and permeabilization.

For cytotoxicity assay, CTLs (after a 3-day activation) continued the activating supplemented with MgCl₂ (1 mM), FeCl₃ (0.05 mM), LDH (Mg²⁺, 1 mM), LDH (Mg²⁺, 1 mM) + FAKi (4 nM), or LDH (Mg²⁺, 1 mM) + NFATi (50 μM), for 4 h. To collect and plate them at different CTL-to-target (B16F10 cells, 10⁴ per well) ratios in the 96-well plate (round bottom) in the presence of 10 ng ml⁻¹ PMA and 0.5 μM ionomycin for 2 h, the percentage of B16F10 cell lysis was quantified using the CytoTox 96 non-radioactive cytotoxicity assay kit. Here, the killing assay was conducted in RPMI 1640 medium with 2% (v/v) of FBS and 1% (v/v) of penicillin-streptomycin at 37 °C in an incubator with 5% (v/v) of carbon dioxide. Since that B16F10 cell line originates from C57BL/6J mouse, allo-reactive BALB/c CTLs could identify the MHC mismatch and cause B16F10 cell death.

To reveal the translocation of nuclear factor of activated T cells (NFAT), NFAT1 level in the cytoplasm and nucleus of CTLs was determined by western blot analysis. First, MgCl₂ (1 mM), FeCl₃ (0.05 mM), LDH (Mg²⁺, 1 mM), FeCl₃ (0.05 mM) + GSH (1 mM), or LDH (Mg²⁺, 1 mM) + GSH (1 mM) was fed in the CTL (2 million per well) activating process for 1 day. After collection, proteins in nucleus and cytoplasm of the treated CTLs were obtained separately using nuclear and cytoplasmic protein extraction kit following the manufacturer's protocol. The both were then diluted with ×4 loading buffer, followed by 10-min boiling. Immunoblots for nuclear fraction and cytoplasmic fraction were performed on PAGE-SDS gels, respectively, followed by detection of NFAT1 monoclonal antibody according to the instructions. The PCNA (proliferating cell nuclear antigen) was used as the loading control for nuclear tests, and β-actin was for cytoplasmic parts.

For IL-2 assay, CTLs (after a 3-day activation; 2 million per well) kept the activating supplemented with FeCl₃ (0.05 mM), LDH (Mg²⁺, 1 mM), or LDH (Mg²⁺, 1 mM) + NFATi (50 μM), for 1 day. Cultural media were collected to quantify IL-2 levels using the mouse IL-2 ELISA kit.

For the assay of glycolytic activity, Seahorse XF96 extracellular flux analyzer (Agilent Technologies) was applied. CTLs (after a 3-day

activation; 0.1 million per well) were seeded onto poly-D-lysine coated XF96 cell plate. XF base medium was recomposed with $MgCl_2$ (1 mM), $FeCl_3$ (0.05 mM), or aPD-1 (5 µg ml$^{-1}$). Reconstituted medium was added into plated CTLs from the beginning of the glycolytic test. Differently, aCD3/28 was applied onto plated CTLs through the multi-injection port of the analyzer, to enable the final well with anti-CD3 antibodies (5 µg ml$^{-1}$) and anti-CD28 antibodies (2.5 µg ml$^{-1}$).

To investigate the PD-1 expression, murine CTLs (after a 3-day activation; 0.2 million per well) were co-cultured with 4T1 cancer cells (0.2 million per well) for different times (6 h; 16 h). After that, all cells were harvested for aPD-1 staining. To facilitate the distinction by flow cytometry, CTLs were pre-labeled with CellTracker Deep Red dye as the manufacturer's instruction.

## Bioluminescence imaging
For all in vivo therapeutic and challenging trials, cancer cells were labeled with luciferase and GFP, to ascertain all cancer cells to be distinguished unambiguously from other cells in the recipient mouse. Therefore, tumor growth and systemic metastasis can be assessed by bioluminescence imaging. Five minutes prior to the imaging, mice were injected intraperitoneally with 100 µl of PBS containing VivoGlo™ Luciferin (5 mg ml$^{-1}$; P1043, Promega). Mice were imaged by an in vivo Xtreme 4MP system (Bruker MI) with the "Luminescence" modality and "Reflectance" background image. For mice euthanized, their visceral organs were collected surgically and imaged, after whole-body imaging. The exposure time ranged from 10 to 60 s, to avoid saturation.

## Single-cell dissociation of primary tissue
Tissues including primary solid tumor, postsurgical wound debris, SLN, and spleen were harvested for single-cell dissociation. Briefly, except for the spleen that avoided the digestion step, other collected tissues were mechanically minced with a sterilized scalpel and then digested in DMEM/F-12 medium, containing 300 U ml$^{-1}$ of collagenase type III, 100 U ml$^{-1}$ of hyaluronidase, 5% (v/v) FBS, 5 µg ml$^{-1}$ of insulin, 10 ng ml$^{-1}$ of epidermal growth factor protein, and 500 ng ml$^{-1}$ of hydrocortisone for 1 h, at 37 °C and 5% (v/v) carbon dioxide. The resultant suspension was then dispersed in DMEM/F-12 medium with 5 mg ml$^{-1}$ of dispase II and 0.1 mg ml$^{-1}$ of DNase I for 5 min at 37 °C. Red blood cell lysis buffer was followed to remove red blood cells within 3 min. The 70-µm cell strainer was used in the final step to collect single-cell suspension.

## GSH and GSSG analysis
GSH and GSSG levels within SLN tissue were analyzed using a Glutathione assay kit. Sample preparation and assay process were conducted by following the recommended protocols provided by the manufacturer strictly.

## Flow cytometric analysis
To identify T-lymphocyte immunomodulation in SLN after flex-patch treatment, SLNs in each group ($n = 5$ per group) were harvested on postsurgical day 5 and day 30, separately, to get single-cell suspension for flow cytometric analyses. LIVE/DEAD™ fixable near-IR dead cell stain kit was applied first to differentiate live and dead cells. Cells were then blocked with 0.25 µg anti-mouse CD16/32 antibodies (aCD16/32) per million cells for 10 min on ice prior to immunostaining. To quantify the population and memory phenotypes of antigen (H-2Ld SPSY-VYHQF)-specific CD8$^+$ T lymphocytes, cells were stained with fluorescence-labeled Pro5 major histocompatibility complex (MHC) class I pentamers, fluorescence-labeled anti-mouse antibodies CD44, CD62L, CD8 and CD3 following the manufacturer's instructions. Fluorescence-labeled Pro5 MHC class I pentamers with H-2Ld MPVGGQSSF were applied for control. For others, cells were stained with fluorescence-labeled anti-mouse antibodies CD3, CD8, Ki67, Granzyme B, CD44, CD62L, CD4, CD25, and Foxp3 following the manufacturer's instructions.

To quantify T leukocytes in postsurgical tumor, single-cell suspension of wound debris in each group was obtained on postsurgical day 5, with LIVE/DEAD staining and aCD16/32 pre-blocking (as the foregoing). Cells were stained with fluorescence-labeled anti-mouse antibodies CD45, CD3, CD8, and CD4 following the manufacturer's instructions.

Here, prior to all intracellular staining steps, cells were fixed and permeabilized with the Foxp3/Transcription Factor Staining Buffer set following the manufacturer's instructions. All stained cells were detected on a CytoFLEX S flow cytometer (Beckman) and quantified by FlowJo software version 10.0.7.

## Blood sera detection
Cytokines including TNF-α, IFN-γ, IL-10, and IL-12p70 in mouse blood sera were measured using ELISA kits, following the manufacturer's instructions. Mouse blood samples in different treatment groups were firstly collected to clot at room temperature. After 30 min, blood samples were centrifuged at 2000 × $g$ for 15 min at 4 °C. The top yellow sera were then collected for cytokines analysis, without disturbing the bottom deposit.

## Statistical analysis
All quantitative data are expressed as the mean ± standard deviation (SD) unless otherwise indicated. For multiple comparisons (more than two groups were compared), one-way ANOVA with Tukey's multiple comparisons was used. For two-group comparisons, unpaired/paired two-tailed $t$-test was used. Mouse survival percentage was compared by a log-rank (Mantel–Cox) test. Statistical analysis was performed using GraphPad Prism 8.0. All $p$ values of <0.05 were considered significant: *$p < 0.05$, **$p < 0.01$, ***$p < 0.001$, and ****$p < 0.0001$.

## Reporting summary
Further information on research design is available in the Nature Portfolio Reporting Summary linked to this article.

## Data availability
The single-cell RNA sequencing data generated in this study have been deposited in the Sequence Read Archive database with the accession code of PRJNA853539. The remaining data are available within the Article, Supplementary Information or Source Data file. Source data are provided with this paper.

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

## Acknowledgements

This work was supported by the National Natural Science Foundation of China (NSFC 32222090, 32101069 and 32171318), the Faculty of Health Sciences, University of Macau, the Multi-Year Research Grant (MYRG) of University of Macau (File no. MYRG2022-00011-FHS), the Science and Technology Development Fund, Macau SAR (File no. 0109/2018/A3, 0011/2019/AKP, 0113/2019/A2, 0103/2021/A, 0092/2020/AMJ, and 0002/2021/AKP), Shenzhen Science and Technology Innovation Commission, Shenzhen-Hong Kong-Macau Science and Technology Plan C

(No. SGDX20201103093600004, and SGDX20201103092601008), and Dr. Stanley Ho Medical Development Foundation (SHMDF-OIRFS/2022/002). We appreciate the assistance and support from the Proteomics, Metabolomics and Drug Development Core, Animal Research Core, and Biological Imaging and Stem Cell Core in the Faculty of Health Sciences, University of Macau.

## Author contributions

B.L. wrote the manuscript. B.L. and G.W. conceived, designed and carried out the majority of experiments. K.M., A.Z., and B.L. performed the single-cell RNA sequencing and data processing. X.Y., L.X., and J.Y. helped animal experiments. L.S. assisted the intravital imaging on LN accumulation of flex-patch and TAMs phagocytosis to cancer cells. J.H.L. carried out the IHC staining and RT-qPCR test. W.L. assisted the in vitro CTL tests. Y.D. and C.-X.D. supervised the research and edited the manuscript. All authors approved and read the final manuscript.

## Competing interests

The patent with the title of the preparation of immuno-adjuvant patch for postsurgical tumor inhibition has been filed in the Chinese Patent Office with the application number 202211113583X, in which Y.D. and B.L. are listed as inventors. The remaining authors declare no other competing interests.
