## [Peer Review File · Nature Communications]

Fueling sentinel node via reshaping cytotoxic T lymphocytes with a flex-patch for post-operative immuno-adjuvant therapyREVIEWER COMMENTS

Reviewer #1 (Remarks to the Author): with expertise in breast cancer, immunotherapy

In this manuscript, the authors proposed an implantable flex-patch for postsurgical immunotherapy, which could reshape T cell phenotypes in immunosuppressive lymph nodes. Subsequently, this device reduces tumor relapse and metastasis. Nevertheless, the mechanism is not clear at this stage. Additional experiments are required to make the conclusions more convincing. Detailed comments are listed as follows:

Major comments:

1. The author showed that LDH nanoadjuvants and CC matrix did not have apparent cellular cytotoxicity on RAW264.7 and 4T1 cell line in vitro in Fig. S2a-d. How about their cytotoxicity in vivo, especially the influence on liver and kidney? Because the data in Fig. S2f showed that the in vivo biodistribution of flex-patch is high in liver and kidney until day24 after operation.
2. The author uses cell tagging technique to monitor SLN metastasis. Additional evidences to prove tumor colonization, such as HE staining, IHC and luciferase luminescence systems are required.
3. The data showed that postsurgical flex-patch implantation could prevent tumor relapse and metastasis. It is recommended to evaluate the proliferation and apoptosis of tumor cells in the metastatic SLN.
4. The experimental replication is problematic. For example, in Figure 3e, the organs from three experimental groups were put into a single dish. Why? The organs from different experiment groups should be presented in separated images. In Figure 3e, n = 1 (group II), n = 3 (group III), n = 5 (group IV). The replications are not enough to get a convincing conclusion.
5. The author showed enhanced proliferation and granzyme B production of CD8+ T cells in group IV (Fig. 4c, d), The tumoricidal experiment is recommended.
6. The author found that glutathion consumption leads to the reduction of Foxp3 expression of CD4+ T cells. What is the underlying mechanism? It has been reported that glutathion abundance in TME influences metabolic activity of CD8+ T cells. Whether glutathion influences CD8+ T cells in your experiment?

Minor comments:

1. There is inconsistent grouping in the manuscript, supplementary Figure 8a for example.
2. For flow cytometric gating strategy, it is recommended to use live/dead staining to exclude dead populations.

Reviewer #2 (Remarks to the Author): with expertise in single-cell genomics and cancer metabolism

In this manuscript entitled "Fueling sentinel node via heterogeneous T-cell immunometabolism reshaping with a flex-patch feeds immunoadjuvant therapy", the authors incorporate magnesium iron-layered double hydroxide (LDH) and anti-PD-1 antibodies (aPD-1) into flex-patch for modulating the immune response in sentinel lymph node (SLN). The observations in this study suggest that LDH may be an effective adjuvant for aPD-1 to boost immune function, however, the exact mechanism of how LDH enhances the effect of aPD-1 on the immune system is not well explained. Whether the adjuvanticity of LDH is through promoting the efficiency of aPD-1 delivery, protecting aPD-1 from degradation during delivery, reshaping cellular metabolism, targeting genes/proteins, or regulating signaling pathways? The authors should comment on this at least.

Other points that the authors should consider are:

1. The authors conclude from GSEA analyses in Suppl. Fig. 10b that aPD-1 downregulates IFN-gamma signaling response in CD8+ T cells, which appears to be contradicted by the observation of other studies that aPD-1 induces IFN-gamma γ production by activated T cells (Ayers et al. 2017. J Clin Invest. PMID: 28650338; Garris et al. 2018. Immunity. PMID: 30552023).
2. The authors should plot the expression of foxp3 and its' target genes in two groups to support their conclusion that "patch decreased cellular Foxp3 genes expression (Fig. 6b)".
3. Usually "scRNA seq" other than "sc-RNA seq" is used as the abbreviation for "single-cell RNA sequencing".
4. Volcano plot should be drawn to exhibit the differences in gene expression of different cell types between two groups.
5. The introduction section misses the motivation to focus only on the metabolic pathway variations. Since (1) the immune responses are linked to diverse factors including signaling processes and metabolism and (2) we don't know whether the effect of LDH on the immune system is via metabolic alterations (Again, the mechanism of how LDH function in the immune system should be discussed in the manuscript), the differences in expression of non-metabolic genes should also be investigated.
6. Please avoid saying "we for the first time identified a correlation between immune dysfunction and aberrant mitochondrial activity." Their correlation has already been reported elsewhere.
7. Study revealed that LDH influences the immune responses by targeting TGF-beta2 (Zhu et al. 2021. ACS Nano. DOI:10.1021/acsnano.0c08727). However, in this study, the authors ascribe the effect of LDH on Foxp3+ Treg cells to TGF-beta1. Could the author please provide more evidence to support their finding and comment on this in the discussion section?
8. The writing could be improved and the grammatical issues should be carefully examined and corrected.

Reviewer #3 (Remarks to the Author): with expertise in nano-engineering, patch systems

In this manuscript, Li et al. designed an immune-fueling flex-patch that was implanted on postoperative wound to deliver immune-invigorating agents into SLN spatiotemporally. The authors validated the antitumor efficacy of this system against orthotopic 4T1 breast tumor model with high spontaneous metastasis. Moreover, the authors found that LDH activated the de novo gene translation of T cells (enriched eukaryotic translation pathways). The goals and results of this research are of particular interest in the field of postsurgical cancer immunotherapy, while the overall study needs well organized and performed. Thus, I suggest publishing after addressing the following issues.

- 1) Please provide the scheme of material design and the corresponding description in the part of the flex-patch fabrication.
- 2) Please provide high-resolution and high-quality SEM images of LDH nanoadjuvant on the flex-patch to meet the standard of publishing.
- 3) Please add the corresponding description of the results of XRD patterns, and also revise the wrong color marks in the figure.
- 4) In Fig. 3e and 3i, one image for several groups make reader confusing. Please provide images individually.
- 5) Please note the ethical endpoint of the animal study. If tumor volume reached 1500–2000 mm³, mice should be killed as per humane animal welfare regulations.
- 6) In the experiments of FACS analysis, regulatory T Cells related to antitumor immunity should be

marked as CD4+CD25+FOXP3+.

7) Other comprehensive experiment of molecular biology should be carried out to support the conclusion that LDH activated the de novo gene translation of T cells.

8) Please provide the data to confirm the in vivo biosafety of the flex-patch.

9) The whole manuscript needs further polishing, including the write logic of the manuscript and the detailed information of the experimental methods.

Reviewer #4 (Remarks to the Author): with expertise in cancer immunology, lymphatic immunobiology

The manuscript by Li et al proposes an interesting strategy to treat the postsurgical wound following tumor resection for the purposes of releasing antibodies and adjuvants to the sentinel lymph node for boosted systemic immune surveillance. The approach is innovative and appears to be effective in the 4T1 model used. While the data presented is intriguing, the mechanistic claims are supported only by very descriptive data and not linked functionally to the therapeutic response. It is therefore unclear to what extent the various changes observed are driving outcomes in order to inform optimization or future use of this technology.

Major Comments:

LDH is never defined, it is unclear what this adjuvant is and why it was chosen.

Figure 2F, it is unclear what is being imaged, assume the inguinal LN basin. Please label the image to facilitate interpretation. An image of an early time point would be more informative, when you expect high doses in the ipsilateral but not contralateral LN. Unclear, why you would still have immunofluorescent tracer at such long time points (5 weeks?) and in distal locations, or what the relevance is for the therapeutic response. Please clarify.

Please provide controls for the PD-1 immunofluorescence in Figure 2.

There are some questions about the number of times the main therapeutic efficacy experiment was performed. Only 6 mice are shown, though the figure legend says the experiment was performed 3 times. Please present all of the data in order to better evaluate the consistency of the therapeutic response. Additionally, therapeutic efficacy is only demonstrated in the 4T1 model, additional models should be employed to demonstrate the broad applicability of this therapy.

It is surprising that 2/5 tumors grew out in the re-challenged mice. A larger n would be helpful here. Can the authors provide data to explain the heterogeneity in this re-challenge experiment?

In discussion of Figure 3, the language used in the text makes it very difficult to follow, please consider revising. For example, was it one mouse in group IV that showed no metastasis? Only the one mouse that had a local recurrence was evaluated?

The latency of the growth of the EMT6 tumors is 150 days? Can the authors provide growth rates in naive mice? Is this an impact of the treatment or do these tumors simply exhibit such poor growth. It is hard to interpret this experiment and accept the claims of the authors when the 4T1 outgrowth in re-challenged mice is so much faster than the antigen-independent model.

The authors make a big point of using Cell tag technology to demonstrate that there is a viable metastasis in the LN at time of resection and treatment, but direct imaging of the LN would be more informative. It is not clear what the overlap in bar codes means in terms of metastatic size or how consistent this is across animals. Please expand and clarify as this is made to be an important assumption in their model.

Please provide more rationale for the use of the pentamer described in the text and antigen target. It

is presumed from the reference this is the immunodominant AH1 epitope. Please confirm. The paper referenced doesn't actually track this response in 4T1 and in fact shows cross-reactivity with CT26 that appears to be in part independent of AH1. Please provide additional references where this pentamer is used to track antigen-specific responses in 4T1 tumors and also provide controls for the pentamer stain (if already in the suppl fig please label more carefully). Please also show the pentamer stain with CD44 to ensure an antigen-experienced phenotype.

While the patch clearly has an effect on tumor recurrence in their 4T1 model, the mechanistic insight into why this is, is very limited. Markers used to define the single cell populations are very limited and the data is only descriptive, there was no functional validation of the proposed metabolic changes observed in either the CD4 or CD8 compartment. More importantly, it is not clear that any of these metabolic changes would even be causal as it is already known that activating T cells will remodel their metabolic state to permit rapid proliferation and effector function. Therefore, whether the differences observed really represent a direct effect of the therapy that then drives more efficient effector function vs a correlate with the activation that is ongoing is unclear. The comparisons made are also only between the combination therapy and unloaded patch, what is the metabolic effect of the individually loaded LDH and PD-1 patches? The Treg data is interesting but with no functional followup it is unclear how this contributes to the therapeutic phenotype observed.

The authors comment on the immune cells that are recruited into the surgical site with and without the patch but make no reference to the rate of healing in these various groups. Can they please provide data on wound closure? Also, while the biology dissected in Fig 7 is likely relevant to the responses observed it is again never functionally linked to the therapeutic response and unclear how or if it contributes.

Throughout the language used is very difficult to interpret, please revise and simplify the language. Words like anatomize, for example, are used to explain simpler concepts of determining metastatic burden, for example. Line 284-287, is another example of text that is difficult to follow

REVIEWER COMMENTS

Reviewer #1 (Remarks to the Author): with expertise in breast cancer, immunotherapy

In this manuscript, the authors proposed an implantable flex-patch for postsurgical immunotherapy, which could reshape T cell phenotypes in immunosuppressive lymph nodes. Subsequently, this devise reduces tumor relapse and metastasis. Nevertheless, the mechanism is not clear at this stage. Additional experiments are require to make the conclusions more convincing. Detailed comments are listed as follows:

Major comments:

1. The author showed that LDH nanoadjuvants and CC matrix did not have apparent cellular cytotoxicity on RAW264.7 and 4T1 cell line in vitro in Fig. S2a-d. How about their cytotoxicity in vivo, especially the influence on liver and kidney? Because the data in Fig. S2f showed that the in vivo biodistribution of flex-patch in high in liver and kidney until day24 after operation.

Response: To assess the in vivo cytotoxicity of our flex-patch, we harvested the murine heart, liver, spleen, lung, and kidney on postsurgical day 36, and observed no obvious histopathological variations in these tissues (Supplementary Figure 6a). To further analyze the biochemical markers in murine sera, the implanted patch barely affected the normal fluctuations of blood urea nitrogen (BUN), creatine kinase (CK), lactate dehydrogenase (LADH), creatinine (CRE), aspartate transaminase (AST), and alanine aminotransferase (ALT) (Supplementary Figure 6b). Our flex-patch exhibits good biocompatibility and biosafety for in vivo application.

Supplementary Figure 6 has been added into the revised Supplementary Information. The corresponding descriptions have been replenished into line 21-27, page 9, in the revised manuscript, and “in vitro and in vivo biosafety of the patch” section in the revised Supplementary Information.

Supplementary Figure 6. In vivo biosafety of the flex-patch. a, Hematoxylin and eosin (H&E) staining of major organs harvested from healthy mice and patch-treated mice on postsurgical day 36. Scale bar: 2.5 mm (low-magnification image); scale bar: 100 μm (high-magnification image). **b**, Heatmap exhibiting the relative biochemical levels in murine sera from healthy group and patch-treated group. BUN, blood urea nitrogen; CK, creatine kinase; LADH, lactate dehydrogenase; CRE, creatinine; AST, aspartate transaminase; ALT, alanine aminotransferase. (n = 3 per group)

2. The author use cell tagging technique to monitor SLN metastasis. Additional evidences to

prove tumor colonization, such as HE staining, IHC and luciferase luminescence systems are required.

Response: Thanks for the suggestion. Apart from the cell tagging technique, we examined the SLN metastasis by labeling 4T1 cancer cells with GFP and luciferase (4T1-GFP-luc). On day 10 post inoculation of 4T1-GFP-luc cells, we collected brachial lymph nodes, axillary lymph nodes, and inguinal lymph nodes from the mouse model and identified the luminescent SLN (Supplementary Figure 2a,b), which resulted from an invasion of neighboring cancer cells. Meanwhile, these infiltrated cancer cells could also be recognized by detecting their strong GFP expression using immunohistochemistry (IHC) analysis (Supplementary Figure 2c).

Supplementary Figure 2. SLN metastatic tracking in postsurgical 4T1 BC mouse model. a, Schematic illustrating the inoculation of 4T1-GFP-luc cancer cells to survey SLN metastasis. **b,** Ex vivo bioluminescence of LNs collected from 4T1-tumor model on post-inoculation day 10. Here, LNs include two brachial LNs, two axillary LNs, and two inguinal LNs. The one next to the 4T1 primary tumor is regarded as SLN, which exhibited bioluminescence. n = 6 mice. **c,** IHC staining for GFP-labeled 4T1 cancer cells infiltrated in SLN. Scale bar: 100 μm. n = 6 mice. **d,**

Schematic illustrating 4T1-celltagging trial to survey SLN metastasis. **e**, 4T1-barcode numbers in SLN on day 10 post inoculation of 4T1-tag cells, quantified by DNA sequencing. n = 10 mice.

Supplementary Figure 2 has been updated in the revised Supporting Information. In addition, the corresponding context has been added in **line 26-30, page 8**, and in Methods of **“Identifying tumor metastasis in SLN”** section in the revised manuscript.

3. The data showed that postsurgical flex-patch implantation could prevent tumor relapse and metastasis. It is recommended to evaluate the proliferation and apoptosis of tumor cells in the metastatic SLN.

Response: On postsurgical day 5, we collected SLNs from group I (untreated) and group V (LDH@aPD-1@CC patch) to evaluate the proliferation and apoptosis of tumor cells through the immunohistochemistry (IHC) analysis. As exhibited in **Fig. 5d**, SLN invasion of 4T1 cancer cells (labeled with GFP) was detected in these two groups. However, different from the high Ki67 (proliferation marker) level in untreated group, SLN in patch-treated mice showed a remarkable Ki67 decrease and cleaved caspase-3 (CC3, apoptosis marker) increase. Intraperitoneal injection of CD8 antibodies (aCD8) in patch-treated mice, we found a recovered high expression of Ki67 (Group V + aCD8, **Fig. 5d**). Thus, flex-patch treatment induced tumoral apoptosis in the metastatic SLN, which associates with of CD8⁺ T cell effector function.

Fig. 5 d, Immunohistochemistry (IHC) staining and quantification of GFP (4T1 marker), Ki67 (proliferation marker), and CC3 (apoptosis marker) in SLN. Scale bar: 100 μ m; n = 3 mice per group. Data are presented as mean \pm SD.

Fig. 5d has been appended in the revised manuscript with the corresponding context in **line 29-30, page 15, and line 1-5, page 16.**

4. The experimental replication is problematic. For example, in Figure 3e, the organs from three experimental groups were put into a single dish. Why? The organs from different experiment groups should be presented in separated images. In Figure 3e, n = 1 (group II), n = 3 (group III), n = 5 (group IV). The replications are not enough to get a convincing conclusion.

Response: We apologize for vague descriptions. In fact, organs in one dish belong to one mouse from a single treatment group. Here, as reviewer suggested, the organs from group I-IV have been presented separately (**Fig. 6g**), to demonstrate the uninhibited postoperative tumor relapse and systemic metastasis.

For n = 1 (group II), it is because of the severe postsurgical recurrence rate for CC (carboxylated chitosan, patch matrix)-treated mice, leaving only one survival for tumor challenge test. To get a convincing conclusion, two breast tumor models including 4T1 and EMT6 were applied with 12 mice per group. Meanwhile, tumor re-challenging trial was carried out in these two models with eight mice per group. As shown in **Fig. 6** and **Supplementary Figure 15**, LDH@aPD-1@CC patch aroused the postsurgical host an efficient and long-term therapeutic defense against BC relapse.

Fig. 6 has been added in the revised manuscript with the corresponding context in **line 4-27, page 19. Supplementary Figure 15** has been shown in the revised Supplementary Information.

Fig. 6 | Postsurgical 4T1-BC treatment of flex-patch. **a**, Schematic illustrating therapeutic procedure on 4T1 BC mouse model after surgery, including no patch implantation (untreated, group I) and patch with different therapeutic agents (CC, group II; aPD-1@CC, group III; LDH@CC, group IV; LDH@aPD-1@CC, group V). n = 12 mice per group. **b**, Representative bioluminescence images of the mice per group. **c,d**, Individual (**c**) and average (**d**) tumor growth kinetics per group. **e,f**, Mice survival (**e**) and weight changes (**f**) per group. **g**, Representative bioluminescence images of visceral organs of tumor mice in group I-IV. **h**, Schematic illustrating the route of 4T1 tumor rechallenge assay. Apart from the control group, all tumor-free mice were harvested from group V (LDH@aPD-1@CC). n = 8 mice per group. **i,j**, Individual (**i**) and average (**j**) tumor growth kinetics per group. **k**, Mice survival in different rechallenge groups. For (**d,f,j**), curve ended when the first mouse in the corresponding group died. Data are presented as mean \pm SD. Statistical significance was calculated via one-way ANOVA with Tukey's multiple comparisons, ****P < 0.0001. For mice survival analysis (**e,k**), statistical significance was calculated via the Log-rank (Mantel-Cox) test, by comparing with the untreated group I, **P < 0.01, ****P < 0.0001.

Supplementary Figure 15. Postsurgical EMT6-BC treatment of flex-patch. **a**, Schematic illustrating therapeutic procedure on EMT6 BC mouse model after surgery, including no patch implantation (untreated, group I) and patch with different therapeutic agents (CC, group II; aPD-1@CC, group III; LDH@CC, group IV; LDH@aPD-1@CC, group V). $n = 12$ mice per group. **b,c**, Individual (**b**) and average (**c**) tumor growth kinetics per group. **d,e**, Mice survival (**d**) and weight changes (**e**) per group. **f**, Schematic illustrating the route of EMT6 tumor rechallenge assay. Apart from the control group, all tumor-free mice were harvested from group V (LDH@aPD-1@CC). $n = 8$ mice per group. **g,h**, Individual (**g**) and average (**h**) tumor growth kinetics per group. **i**, Mice

survival in different rechallenge groups. For (c,e,h), curve ended when the first mouse in the corresponding group died. Data are presented as mean \pm SD. Statistical significance was calculated via one-way ANOVA with Tukey's multiple comparisons, ****P < 0.0001. For mice survival analysis (d,i), unless indicated otherwise, statistical significance was calculated via the Log-rank (Mantel-Cox) test, by comparing with the untreated group I, ***P < 0.001, ****P < 0.0001.

5. The author showed enhanced proliferation and granzyme B production of CD8+ T cells in group IV (Fig. 4c, d), The tumoricidal experiment is recommended.

Response: For your information, since we added a new group of LDH@CC for in vivo therapeutic experiment, the previous group IV (LDH@aPD-1@CC) has become group V in the revised manuscript. Now, five groups are listed as group I (untreated), group II (CC), group III (aPD-1@CC), group IV (LDH@CC), and group V (LDH@aPD-1@CC).

On day 5 after the patch implantation, we identified speedy increases of proliferation (Ki67⁺) and killing effect (indicated by the expression of cytotoxic granzyme B (GzmB⁺)) of CD8⁺ T cells in the metastatic SLN (Fig. 5b,c). LDH adjuvant amplified the functional effect at about two-fold more than aPD-1 did (group IV versus group III, Fig. 5b,c), indicating a promoting performance of LDH for cytotoxic T lymphocyte (CTL) effector function. To confirm it, we co-cultured BALB/c CTLs with B16F10 cancer cells for a 2-h killing assay. Here, BALB/c CTLs could induce B16F10 cell death because of the MHC mismatch (*Nat. Commun.*, 2017, 8, 511; *Science*, 2021, 374, 299), and LDH adjuvant empowered these CTLs a stronger killing capacity (Fig. 4d). It is because that functional metal cations (Mg²⁺ and Fe³⁺) from LDH participate in the modulation of T cell receptor (TCR) signaling of CTLs and promote their activation and cytotoxicity, comprehensively demonstrated in the section of “LDH adjuvant promotes the activation and cytotoxicity of CTLs” in the revised manuscript.

Fig. 5 b,c, Flow cytometric quantification of Ki67⁺CD8⁺ T cells (**b**) and GzmB⁺CD8⁺ T cells (**c**), gating on CD8⁺CD3⁺ cells in SLN. n = 5 mice per group. Data are presented as mean ± SD. Statistical significance was calculated via one-way ANOVA with Tukey's multiple comparisons. *P < 0.05, **P < 0.01, ***P < 0.001, ****P < 0.0001. **d**, Immunohistochemistry (IHC) staining and quantification of GFP (4T1 marker), Ki67 (proliferation marker), and CC3 (apoptosis marker) in SLN. Scale bar: 100 μm; n = 3 mice per group. Data are presented as mean ± SD.

Fig. 4 d, A 2-h killing assay of CTLs, varying BALB/c CTLs to targets (B16F10 cancer cells) ratio.

Moreover, to examine the treated SLN and postsurgical tumor residual via IHC analyses, high intensity of cleaved caspase-3 (CC3, apoptosis marker) was observed (Fig. 5d and Fig. 71), which however became unobvious with the injection of CD8 antibodies (aCD8). Flex-patch treatment induced tumoricidal effect, which associates with enhanced effector function of CD8⁺ T cells.

Fig. 4d, Fig. 5b-d and Fig. 71 have been replenished into the revised manuscript with the corresponding descriptions in line 10-16, page 13, line 29-30, page 15, line 1-5, page 16, and line 15-17, page 22.

Fig. 7 I, IHC detection and quantification of 4T1 tumoral proliferation and apoptosis (GFP, 4T1 marker; Ki67, cell proliferation marker; CC3, cell apoptosis marker) in postsurgical tumor residual. Scale bar: 100 µm; n = 3 mice per group. Data are presented as mean ± SD.

6. The author found that glutathione consumption leads to the reduction of Foxp3 expression of CD4⁺ T cells. What is the underlying mechanism? It has been reported that glutathione abundance in TME influences metabolic activity of CD8⁺ T cells. Whether glutathione influences CD8⁺ T cells in your experiment?

Response: For CD4⁺ T cells, a previous study (*Cell Metab.*, 2020, 31, 920) revealed that high serine metabolism downregulated Foxp3 expression of Treg cells, and this serine metabolism was

restricted by glutathione (GSH), to preserve Treg functionality. Downregulating GSH in Treg cells therefore reduced their Foxp3 and their suppressive capacity. Consistent with this report, our immune adjuvant LDH generated reactive oxygen species (ROS; **Supplementary Figure 14b**), the GSH scavenger, which consumed GSH (**Fig. 5k,l**) and finally modulated Foxp3 levels in CD4⁺ T cells (**Fig. 5m**).

Supplementary Figure 14. b, Representative histograms and quantifications of the intracellular ROS levels in CD8⁺ T cells and CD4⁺ T cells from SLN. n = 5 mice per group.

Fig. 5 k,l, GSH concentration (**k**) and GSSG concentration (**l**) of SLN tissue on postsurgical day 5. **m**, Flow cytometric quantifications of Foxp3⁺CD25⁺CD4⁺ T cells, gating on CD4⁺CD3⁺ cells

in SLN on postsurgical day 5. n = 5 mice per group. Data are presented as mean \pm SD. Statistical significance was calculated via one-way ANOVA with Tukey's multiple comparisons. *P < 0.05, **P < 0.01, ***P < 0.001, ****P < 0.0001.

Additionally, as reviewer reminded, cellular ROS participates in the modulation of T-cell receptor signaling of cytotoxic T lymphocytes (CTLs) via a nuclear factor of activated T-cells (NFAT)-related pathway (*Trends Immunol.*, 2018, 39, 489; *Immunity*, 2013, 38, 225; *Nat. Immunol.*, 2004, 5, 818). To stimulate CTLs with ROS-promoter Fe³⁺ or Fe³⁺-involving LDH, we identified the translocation of NFAT1 from the cytoplasm to the nucleus (Fig. 4g), suggesting that NFAT was activated and performed as a transcription factor. The nuclear import of NFAT1 was downregulated while adding ROS-consumer (glutathione, GSH), which confirmed the ROS-relevant function of Fe³⁺ on NFAT activation. Inhibiting the activation with NFAT inhibitor (NFATi; 11R-VIVIT TFA) interfered the downstream IL-2 induction (Fig. 4h) and the contribution of LDH on CTL activation and cytotoxic killing (Fig. 4a-d). GSH consumption could program CTL immunity.

Fig. 4a-d,g,h and Fig. 5k-m have been added into the revised manuscript with the corresponding descriptions in line 13-22, page 15 and line 21-29, page 18. Supplementary Figure 14 has been shown in the revised Supplementary Information.

Fig. 4 a-c, Representative histograms and quantifications of the expression of early activation marker CD69 (**a**), activation marker CD25 (**b**), and degranulation marker CD107a (**c**) on murine CTLs. **d**, a 2-h killing assay of CTLs, varying BALB/c CTLs to targets (B16F10 cancer cells) ratio. **g**, NFAT1 (also known as NFATc2, nuclear factor of activated T cells, cytoplasmic 2) levels in nuclear and cytoplasmic fraction, investigated by western immunoblot. **h**, CTL IL-2 levels in the cellular media. For all tests, n = 3 per group. For **a-d** and **h**, data are presented as mean ± SD, and statistical significance was calculated via one-way ANOVA with Tukey's multiple comparisons. *P < 0.05, **P < 0.01, ***P < 0.001, ****P < 0.0001.

Minor comments:

1. There is inconsistent grouping in the manuscript, supplementary Figure 8a for example.

Response: We apologize for inconsistent groups in the manuscript, since that LDH@CC group was investigated in scRNA seq trial, but not in in vivo therapy. Now, group I (untreated), group II (CC), group III (aPD-1@CC), group IV (LDH@CC), and group V (LDH@aPD-1@CC) have been evaluated on two breast tumor models (Fig. 6 and Supplementary Figure 15) in the revised manuscript.

2. For flow cytometric gating strategy, it is recommended to use live/dead staining to exclude dead populations.

Response: As suggested, live/dead staining has been applied for flow cytometric gating including Supplementary Figure 11, 16, and 18.

Supplementary Figure 11. Gating strategy in SLN. Representative flow cytometric plots showing the gating strategy to identify CD8⁺, CD4⁺ subsets in SLN.

Supplementary Figure 16. Gating strategy in postsurgical wound. Representative flow cytometric plots showing the gating strategy used to identify CD45⁺ leucocytes and CD45⁺ subsets in postsurgical wound.

Supplementary Figure 18. TAMs gating strategy. Representative flow cytometric plots showing the gating strategy used to identify CD80, CD86, CD206, and PD-1 expressions on TAMs.

Reviewer #2 (Remarks to the Author): with expertise in single-cell genomics and cancer metabolism

In this manuscript entitled “Fueling sentinel node via heterogeneous T-cell immunometabolism reshaping with a flex-patch feeds immunoadjuvant therapy”, the authors incorporate magnesium iron-layered double hydroxide (LDH) and anti-PD-1 antibodies (aPD-1) into flex-patch for modulating the immune response in sentinel lymph node (SLN). The observations in this study suggest that LDH may be an effective adjuvant

for aPD-1 to boost immune function, however, the exact mechanism of how LDH enhances the effect of aPD-1 on the immune system is not well explained.

1. Whether the adjuvanticity of LDH is through promoting the efficiency of aPD-1 delivery, protecting aPD-1 from degradation during delivery, reshaping cellular metabolism, targeting genes/proteins, or regulating signaling pathways? The authors should comment on this at least.

Response: Thank you so much for this constructive suggestion. It is well known that aPD-1 save the effector function of cytotoxic T lymphocytes (CTLs) to some extent (*Immunity*, 2016, 45, 358; *Sci. Adv.*, 2020, 6, eabd2712), demonstrated by improved T-cell receptor (TCR) signaling (*VPS37B*), effector function (*IFRD1*), survival (*HSPA5*, *MAP1LC3B*) (*Gene*, 2017, 618, 14; *Nat. Immunol.*, 2014, 15, 1152) from our scRNA seq trial (**Fig. 3c**), and upregulated glycolytic activity from the glycolysis stress test (**Fig. 3g,h**).

Fig. 3 c, Volcano plots showing DEGs in activated CD8⁺ T cells between aPD-1@CC versus (vs.) CC. Significant DEGs (blue and red dots): p value adjusted < 0.05 and fold change > 2. **g,h**, Glycolytic activity of murine CTLs with diverse treatments. Glycolytic activity (ΔECAR) is

determined by subtracting ECAR maximum from baseline. n = 3 per group. Data are presented as mean \pm SD. Statistical significance was calculated via one-way ANOVA with Tukey's multiple comparisons. *P < 0.05, **P < 0.01, ***P < 0.001.

Intriguingly, the existence of LDH adjuvant promotes CTL activation and cytotoxicity via metal cation-mediated shaping, after a comprehensive investigation:

Two functional metal cations (Mg^{2+} and Fe^{3+} ions) from the LDH adjuvant were separately analyzed here. As exhibited in Fig. 4a-c, in the aCD3/28-activating process (aCD3/28, anti-CD3 and anti-CD28 antibodies), either Mg^{2+} or Fe^{3+} enhanced the expression of CD69 (early activation marker), CD25 (activation marker), and CD107a (degranulation marker) of CTLs. To utilize the MHC mismatch between BALB/c CTLs and B16F10 cancer cells for a cytotoxic test (*Nat. Commun.*, 2017, 8, 511; *Science*, 2021, 374, 299), treated CTLs performed an enhanced killing capacity to targets (B16F10 cancer cells), especially when the population of CTLs was limited. In a 2-h killing assay (Fig. 4d), LDH-modulated CTLs killed ~20% of targets at a CTL-to-target ratio of 0.62:1. By contrast, CTLs from the control group nearly failed to work. It suggested a positive function on CTL activation and cytotoxicity, from either metal cation of LDH adjuvant.

Mg^{2+} ions have been reported to be required by cell-surface molecule lymphocyte function-associated antigen 1 (LFA-1) for an active confirmation on $CD8^+$ T cells, boosting TCR signaling transduction, metabolic programming, and even cytotoxicity (*Cell*, 2022, 185, 585). LFA-1-driven signaling activity including proximal focal adhesion kinase (FAK) and distal extracellular signal-regulated kinase 1/2 (ERK1/2) thus could behave in an Mg^{2+} -modulated manner. As indicated in Fig. 4e,f, phosphorylated FAK (pFAK) and ERK1/2 (pERK1/2) were augmented under the stimulation of free Mg^{2+} or Mg^{2+} -containing LDH, but not Fe^{3+} . Blocking the regulation with FAK inhibitor (FAKi; PF-562271) reduced the distal pERK1/2 (Fig. 4f), and ultimately CTL activation and killing performance (Fig. 4a-d). Mg^{2+} cations from LDH adjuvant could shape CTL immunity successfully (Fig. 4i).

Fig. 4 | In vitro activation and cytotoxicity of CTLs regulated by LDH adjuvant. a-c, Representative histograms and quantifications of the expression of early activation marker CD69 (a), activation marker CD25 (b), and degranulation marker CD107a (c) on murine CTLs. **d,** a 2-h killing assay of CTLs, varying BALB/c CTLs to targets (B16F10 cancer cells) ratio. **e,f,** Representative histograms and quantifications of the phosphorylation of FAK (focal adhesion kinase; e) and the phosphorylation of ERK1/2 (f); **g,** NFAT1 (also known as NFATc2, nuclear factor of activated T cells, cytoplasmic 2) levels in nuclear and cytoplasmic fraction, investigated by western immunoblot. **h,** CTL IL-2 levels in the cellular media. **i,** Representative pathways involved in CTLs immune shaping by LDH adjuvant. For all tests, n = 3 per group. For **a-f** and **h,** data are presented as mean \pm SD, and statistical significance was calculated via one-way ANOVA with Tukey's multiple comparisons. *P < 0.05, **P < 0.01, ***P < 0.001, ****P < 0.0001.

Moreover, cellular ROS participates in the modulation of TCR signaling of CTLs via a nuclear factor of activated T-cells (NFAT)-related pathway (*Trends Immunol.*, 2018, 39, 489; *Immunity*, 2013, 38, 225; *Nat. Immunol.*, 2004, 5, 818). To stimulate CTLs with ROS-promoter Fe³⁺ or Fe³⁺-involving LDH, we identified the translocation of NFAT1 from the cytoplasm to the nucleus (Fig. 4g), suggesting that NFAT was activated and performed as a transcription factor. The nuclear import of NFAT1 was not obviously found in Mg²⁺-treated group, but was downregulated while adding ROS-consumer (glutathione, GSH), which confirms the ROS-relevant function of Fe³⁺ on NFAT activation. Inhibiting the activation with NFAT inhibitor (NFATi; 11R-VIVIT TFA) interfered the downstream IL-2 induction (Fig. 4h) and the contribution of LDH on CTL effector function (Fig. 4a-d). Taken together, Fe³⁺ cations from LDH adjuvant program CTL immunity through a ROS-NFAT axis (Fig. 4i).

Fig. 3c,g,h and Fig. 4 have been added in the revised manuscript with the corresponding descriptions in line 20-28, page 12, line 5-26, page 13, and line 13-22, page 15.

Other points that the authors should consider are:

2. The authors conclude from GSEA analyses in Suppl. Fig. 10b that aPD-1 downregulates IFN-gamma signaling response in CD8+ T cells, which appears to be contradicted by the

observation of other studies that aPD-1 induces IFN-gamma production by activated T cells (Ayers et al. 2017. J Clin Invest. PMID: 28650338; Garris et al. 2018. Immunity. PMID: 30552023).

Response: From Previous Supplementary Figure 9a, the reviewer may notice that we identified two CD8⁺ T subclusters including “Activated CD8⁺ T cells” and “CD8⁺ T cells with IFN expressing”. In the cluster of “Activated CD8⁺ T cells”, no downregulated IFN signaling was enriched after aPD-1 administration. To investigate their aPD-1-mediated IFN-γ expression, we prepared activated CD8⁺ T cells (CTLs) in “metastatic SLN”-like environment, by stimulating murine CD8⁺ T cells with aCD3/28 (anti-CD3 and CD28 antibodies) and co-culturing them with 4T1 cancer cells up to 16 h. Collected CTLs exhibited an obvious rising on surface PD-1 expression (Supplementary Figure 4a). To treat them with aPD-1, we demonstrated an increased IFN-γ level (Supplementary Figure 4b), in consistent with the studies the reviewer mentioned.

Previous Supplementary Figure 9. a, Expression heatmap of cell-type-specific genes in two identified CD8⁺ T subclusters. Columns represent different subclusters and rows represent signature genes.

Supplementary Figure 4. a, PD-1 expression of murine CTLs pre-activated by aCD3/28 and 4T1 cancer cells. **b**, IFN- γ intensity of murine CTLs with and without aPD-1 treatment. n = 3 per group. Data are presented as mean \pm SD.

However, in the cluster of “CD8⁺ T cells with IFN expressing” (Previous Supplementary Figure 10b), we observed that aPD-1 downregulated IFN- γ signaling. One speculation is that it associates with T-cell terminal exhaustion. Terminally exhausted T cells represent defects in the production of IFN- γ (*J. Immunol.*, 2010, 185, 3643; *Proc. Natl. Acad. Sci. U.S.A.*, 2007, 104, 4565), and resist to be reinvigorated by PD-1 blockade treatment (*Science*, 2016, 354, 1160; *Science*, 2016, 354, 1165; *Nat. Immunol.*, 2013, 14, 603). Here, based on the uncertainty, we deleted the content on “CD8⁺ T cells with IFN expressing” cluster in the revised manuscript.

b

Representative pathways enriched in CD8⁺ T cells with IFN expressing

Previous Supplementary Figure 10. b, Representative GSEA-enriched pathways for CD8⁺ T cells with IFN expressing in LDH@CC vs. CC and aPD-1@CC vs. CC. Here p value represents the familywise-error rate (FWER) p value; NES, normalized enrichment score; FDR, false discovery rate.

3. The authors should plot the expression of foxp3 and its' target genes in two groups to support their conclusion that “patch decreased cellular Foxp3 genes expression (Fig. 6b)”.

Response: As exhibited in **Supplementary Figure 15c**, LDH@aPD-1@CC patch decreased cellular Foxp3 genes expression in our scRNA seq trial. This figure has been added in the revised Supplementary Information with the corresponding content in **line 1-2, page 19**, in the revised manuscript.

Supplementary Figure 15. Analyses of Treg cells by scRNA seq. a,b, Expression heatmap of cell-type-specific genes in five CD4⁺ T subclusters (a) with their UMAP plot (b). Columns represent different subclusters and rows represent signature genes. **c**, Foxp3 expression level in Treg cells. Statistical significance was calculated via unpaired two-tailed t-test. ***P < 0.001.

4. Usually “scRNA seq” other than “sc-RNA seq” is used as the abbreviation for “single-cell RNA sequencing”.

Response: Thanks for the advice. We have modified the abbreviation for “single-cell RNA sequencing” into ‘scRNA seq’.

5. Volcano plot should be drawn to exhibit the differences in gene expression of different cell types between two groups.

Response: Thanks for the suggestion. In identified CD8⁺ T subsets, activated CD8⁺ T cells were highlighted (Fig. 3b) to investigate their differentially expressed genes (DEGs). Distinct from CC treatment, activated CD8⁺ T cells after aPD-1@CC therapy (Fig. 3c) exhibited the up-regulation of TCR signaling (*VPS37B*), effector function (*IFRD1*), and survival (*HSPA5*, *MAP1LC3B*) (*Gene*, 2017, 618, 14; *Nat. Immunol.*, 2014, 15, 1152). Similar improvements involving TCR signaling (*VPS37B*), activation (*AMD1*, *IRS2*), effector function (*IFRD1*), proliferation (*DUSP10*), and persistence (*FOSL2*), were also pronounced (Fig. 3d) after LDH@CC management (*Immunity*, 2011, 35, 871; *J. Biol. Chem.*, 1995, 270, 28527; *Nature*, 2004, 430, 793; *Cell Rep.*, 2018, 23, 2142).

Fig. 3b-d has been replenished in the revise manuscript with the corresponding context in line 11-17, page 10.

Fig. 3 b, UMAP plot highlighting the identified activated CD8⁺ T cells. **c,d**, Volcano plots showing DEGs in activated CD8⁺ T cells between aPD-1@CC versus (vs.) CC (**c**) and LDH@CC vs. CC (**d**). Significant DEGs (blue and red dots): p value adjusted < 0.05 and fold change > 2.

6. The introduction section misses the motivation to focus only on the metabolic pathway variations. Since (1) the immune responses are linked to diverse factors including signaling processes and metabolism and (2) we don't know whether the effect of LDH on the immune system is via metabolic alterations (Again, the mechanism of how LDH function in the immune system should be discussed in the manuscript), the differences in expression of non-metabolic genes should also be investigated.

Response: We totally agree that immune responses link to diverse factors, apart from metabolism. Thus, as the reviewer suggested in the beginning, a comprehensive investigation on LDH function has been carried out to reveal that this adjuvant promotes the activation and cytotoxicity of cytotoxic T lymphocytes (CTLs) via metal cation-mediated shaping, detailed in the foregoing response #1.

Briefly, Mg²⁺ ions are required by cell-surface molecule lymphocyte function-associated antigen 1 (LFA-1) for an active confirmation on CD8⁺ T cells (*Cell*, 2022, 185, 585). Mg²⁺ cations from LDH adjuvant could regulate LFA-1-driven signaling activity including proximal focal adhesion kinase (FAK) and distal extracellular signal-regulated kinase 1/2 (ERK1/2), boosting TCR signaling transduction, metabolic programming, and even cytotoxicity (**Fig. 4i**).

Additionally, cellular ROS participates in the modulation of TCR signaling of CTLs via a nuclear factor of activated T-cells (NFAT)-related pathway (*Trends Immunol.*, 2018, 39, 489; *Immunity*, 2013, 38, 225; *Nat. Immunol.*, 2004, 5, 818). To stimulate CTLs with ROS-promoter Fe³⁺, we identified that Fe³⁺-involving LDH adjuvant programmed CTL immunity through a ROS-NFAT axis (**Fig. 4i**).

Fig. 4 i, Representative pathways involved in CTLs immune shaping by LDH adjuvant.

“Immunometabolism” in the manuscript title has been deleted, and **Introduction** section has been modified in the revised manuscript.

7. Please avoid saying “we for the first time identified a correlation between immune dysfunction and aberrant mitochondrial activity.” Their correlation has already been reported elsewhere.

Response: The inappropriate expressions such as “for the first time” have been removed as the reviewer’s kind advice.

8. Study revealed that LDH influences the immune responses by targeting TGF-beta2 (Zhu et al. 2021. ACS Nano. DOI:10.1021/acsnano.0c08727). However, in this study, the authors ascribe the effect of LDH on Foxp3+ Treg cells to TGF-beta1. Could the author please provide more evidence to support their finding and comment on this in the discussion section?

Response: In the reference of *ACS Nano* (2021, 15, 2812), LDH that the authors studied is Magnesium Aluminum layered double hydroxide (MgAl-LDH). In our case, LDH we used is Magnesium Iron layered double hydroxide (MgFe-LDH; **Fig. 2a-c**). Given the existence of reactive oxygen species (ROS)-promoter Fe^{3+} , MgFe-LDH adjuvant upregulated the intracellular ROS level of CD4^+ T cells including Treg cells (**Supplementary Figure 14b**). These generated ROS associate with TGF- β 1 activating based on previous studies (*Lab. Invest.*, 2004, 84, 1013; *Mol. Endocrinol.*, 1996, 10, 1077). In the revised manuscript, we withdrew TGF- β 1-related information since it has no correlation with the theme in this study, about how LDH cooperates aPD-1 on enhancing the effector function of cytotoxic T lymphocytes (CTLs) in metastatic SLN.

Fig. 2 a, Representative transmission electron microscopy (TEM) image of LDH nanosheets (scale bar: 50 nm). **b**, Photograph and schematic contents of the flex-patch. **c**, Representative scanning electron microscopy (SEM) image of the flex-patch with elemental mapping (scale bar: 2 μm). Uniform distribution of aPD-1 labelled with PE fluorophore was imaged by fluorescence microscopy (scale bar: 20 μm).

Supplementary Figure 14. ROS-relevant characterizations of patch-treated SLNs. SLNs in each group were harvested on postsurgical day 5, respectively. **a**, Schematic illustrating the SLN survey on 4T1 BC mouse model after surgery, including no flex-patch implantation (untreated, group I) and flex-patch with different therapeutic agents (CC, group II; aPD-1@CC, group III; LDH@CC, group IV; LDH@aPD-1@CC, group V). **b**, Representative histograms and quantifications of the intracellular ROS levels in CD8⁺ T cells and CD4⁺ T cells from SLN. **c**, Representative flow cytometric plots of Foxp3⁺CD25⁺CD4⁺ T cells, gating on CD4⁺CD3⁺ cells. n = 5 mice per group.

9. The writing could be improved and the grammatical issues should be carefully examined and corrected.

Response: Thanks. Writing has been carefully polished with corrected grammar.

Reviewer #3 (Remarks to the Author): with expertise in nano-engineering, patch systems

In this manuscript, Li et al. designed an immune-fueling flex-patch that was implanted on postoperative wound to deliver immune-invigorating agents into SLN spatiotemporally. The authors validated the antitumor efficacy of this system against orthotopic 4T1 breast tumor model with high spontaneous metastasis. Moreover, the authors found that LDH activated the de novo gene translation of T cells (enriched eukaryotic translation pathways). The goals and results of this research are of particular interest in the field of postsurgical cancer immunotherapy, while the overall study needs well organized and performed. Thus, I suggest publishing after addressing the following issues.

Response: We appreciate the highly positive comment from the reviewer.

1) Please provide the scheme of material design and the corresponding description in the part of the flex-patch fabrication.

Response: As suggested, a schematic illustration of the flex-patch design has been prepared in the revised Supplementary Figure 1a.

In fact, details on the flex-patch fabrication were provided in “Methods” section. LDH nanoadjuvants were firstly synthesized via a one-step co-precipitation. The intrinsic property of LDH enables itself a positive charge (30.7 ± 1.1 mV). Given rich amino groups of aPD-1, carboxylated chitosan (CC; -34.0 ± 6.8 mV) assembles with LDH and aPD-1 through an electrostatic interaction easily. After integrating these three key components ultrasonically, we dropped the mixture on cell cultural lid. The flex-patch was thus formed in a gentle desiccating progress. Here, fabrication details of the flex-patch has been replenished in line 9-13, page 6, in the revised manuscript.

Supplementary Figure 1a. The schematic illustration of the flex-patch design.

2) Please provide high-resolution and high-quality SEM images of LDH nanoadjuvant on the flex-patch to meet the standard of publishing.

Response: As suggested, we have updated a high-quality SEM image to exhibit a rough surface of the flex-patch in the revised Fig. 2c. Given that SEM cannot capture the morphology of small-size LDH with high resolution, TEM characterization has been applied to present these nanoparticles with a sheet-like structure in the revised Fig. 2a. Fig. 2a,c have been added in the revised manuscript.

Fig. 2 a, Representative transmission electron microscopy (TEM) image of LDH nanosheets (scale bar: 50 nm). **b,** Photograph and schematic contents of the flex-patch. **c,** Representative scanning

electron microscopy (SEM) image of the flex-patch with elemental mapping (scale bar: 2 μm). Uniform distribution of aPD-1 labelled with PE fluorophore was imaged by fluorescence microscopy (scale bar: 20 μm).

3) Please add the corresponding description of the results of XRD patterns, and also revise the wrong color marks in the figure.

Response: We apologize that no sufficient description was provided so that a misunderstanding on the results of XRD patterns. LDH nanosheets showed their typical XRD pattern with (003), (006), and (110) peaks (LDH in Supplementary Figure 1c). To collect nanoparticles released from the flex-patch, they showed similar XRD pattern as that of LDH, demonstrating a successful disassembling of the LDH@aPD-1@CC flex-patch (released nanoparticles in Supplementary Figure 1c).

However, to insert fluorophore Cy5.5 into LDH interlayer (LDH-Cy5.5) for an easy intravital tracking, space between LDH host layers would be enlarged, leading to a left shift of (003) and (006) peaks (LDH-Cy5.5 in Supplementary Figure 1c; *Angew. Chem. Int. Ed.* 2000, 39, 22; *Adv. Funct. Mater.* 2020, 30, 1909745). Here, the left shift in LDH-Cy5.5 predominated after we integrated its XRD pattern with that of LDH into one figure.

As suggested, the corresponding descriptions for the results of XRD patterns have been added in line 3-6 and line 21, page 6, and line 13-14, page 9 in the revised manuscript.

Supplementary Figure 1c. XRD patterns of different samples.

4) In Fig. 3e and 3i, one image for several groups make reader confusing. Please provide images individually.

Response: We apologize for the confusion we caused, and individual images have been shown in the revised **Fig. 6g**.

Fig. 6 g, Representative bioluminescence images of visceral organs of tumor mice in group I-IV. Group I, untreated; group II, CC; group III, aPD-1@CC; group IV, LDH@CC; group V, LDH@aPD-1@CC.

5) Please note the ethical endpoint of the animal study. If tumor volume reached 1500–2000 mm³, mice should be killed as per humane animal welfare regulations.

Response: As suggested, two breast tumor models (4T1 and EMT6) have been established to demonstrate the therapeutic effect of our flex-patch in the revised manuscript. In the course of monitoring, mice were killed when tumor volume reached 1000 mm³ (**Fig. 6** and **Supplementary Figure 15**).

Fig. 6 | Postsurgical 4T1-BC treatment of flex-patch. **a**, Schematic illustrating therapeutic procedure on 4T1 BC mouse model after surgery, including no patch implantation (untreated, group I) and patch with different therapeutic agents (CC, group II; aPD-1@CC, group III; LDH@CC, group IV; LDH@aPD-1@CC, group V). n = 12 mice per group. **b**, Representative bioluminescence images of the mice per group. **c,d**, Individual (**c**) and average (**d**) tumor growth kinetics per group. **e,f**, Mice survival (**e**) and weight changes (**f**) per group. **g**, Representative bioluminescence images of visceral organs of tumor mice in group I-IV. **h**, Schematic illustrating the route of 4T1 tumor rechallenge assay. Apart from the control group, all tumor-free mice were harvested from group V (LDH@aPD-1@CC). n = 8 mice per group. **i,j**, Individual (**i**) and average (**j**) tumor growth kinetics per group. **k**, Mice survival in different rechallenge groups. For (**d,f,j**), curve ended when the first mouse in the corresponding group died. Data are presented as mean \pm SD. Statistical significance was calculated via one-way ANOVA with Tukey's multiple comparisons, ****P < 0.0001. For mice survival analysis (**e,k**), statistical significance was calculated via the Log-rank (Mantel-Cox) test, by comparing with the untreated group I, **P < 0.01, ****P < 0.0001.

Supplementary Figure 15. Postsurgical EMT6-BC treatment of flex-patch. **a**, Schematic illustrating therapeutic procedure on EMT6 BC mouse model after surgery, including no patch implantation (untreated, group I) and patch with different therapeutic agents (CC, group II; aPD-1@CC, group III; LDH@CC, group IV; LDH@aPD-1@CC, group V). n = 12 mice per group. **b,c**, Individual (**b**) and average (**c**) tumor growth kinetics per group. **d,e**, Mice survival (**d**) and weight changes (**e**) per group. **f**, Schematic illustrating the route of EMT6 tumor rechallenge assay. Apart from the control group, all tumor-free mice were harvested from group V (LDH@aPD-1@CC). n = 8 mice per group. **g,h**, Individual (**g**) and average (**h**) tumor growth kinetics per group. **i**, Mice

survival in different rechallenge groups. For (c,e,h), curve ended when the first mouse in the corresponding group died. Data are presented as mean \pm SD. Statistical significance was calculated via one-way ANOVA with Tukey's multiple comparisons, ****P < 0.0001. For mice survival analysis (d,i), unless indicated otherwise, statistical significance was calculated via the Log-rank (Mantel-Cox) test, by comparing with the untreated group I, ***P < 0.001, ****P < 0.0001.

6) In the experiments of FACS analysis, regulatory T Cells related to antitumor immunity should be marked as CD4⁺CD25⁺FOXP3⁺.

Response: As suggested, regulatory T cells have been gated by CD4⁺CD25⁺Foxp3⁺ in the revised Supplementary Figure 11.

Supplementary Figure 11. Gating strategy in SLN. Representative flow cytometric plots showing the gating strategy to identify CD8⁺, CD4⁺ subsets in SLN.

7) Other comprehensive experiment of molecular biology should be carried out to support the conclusion that LDH activated the de novo gene translation of T cells.

Response: Via the gene set enrichment analysis (GSEA), we identified an obvious contribution of LDH adjuvant on the initiation of eukaryotic translation (Fig. 3i) of activated CD8⁺ T cells (shorted as CTLs). To validate the expression of top 10-ranked genes from this enrichment by quantitative real-time polymerase chain reaction (RT-qPCR), increased transcription of EIF5, RPL5, RPS6, RPL7, RPS15, RPS14, RPL17, RPL15, RPL13A, and RPL18A was found in LDH@CC-treated CTLs, compared to CC group (Fig. 3j). Our data indicated a remarkable influence of LDH adjuvant to CTL biology.

Fig. 3 i, Representative GSEA-enriched pathways for activated CD8⁺ T cells, in LDH@CC vs. CC. GSEA p value represents the familywise-error rate (FWER) p value; NES, normalized enrichment score; FDR, false discovery rate. n = 6 per group. Significantly enriched pathways with GSEA FWER p value < 0.05. **j**, RT-qPCR validation of scRNA seq data for enriched top 10-ranked genes from (i) in murine CTLs. Relative quantitation was obtained by normalization to 18S expression. n = 3 per group. Data are presented as mean ± SD. Statistical significance was calculated via paired two-tailed t-test. *P < 0.05, **P < 0.01, ***P < 0.001, and ****P < 0.0001.

To explore further, LDH adjuvant promotes CTL activation and cytotoxicity via metal cation-mediated programming:

Two functional metal cations (Mg^{2+} and Fe^{3+} ions) from the LDH adjuvant were separately analyzed here. As exhibited in **Fig. 4a-c**, in the aCD3/28-activating process (aCD3/28, anti-CD3 and anti-CD28 antibodies), either Mg^{2+} or Fe^{3+} enhanced the expression of CD69 (early activation marker), CD25 (activation marker), and CD107a (degranulation marker) of CTLs. To utilize the MHC mismatch between BALB/c CTLs and B16F10 cancer cells for a cytotoxic test (*Nat. Commun.*, 2017, 8, 511; *Science*, 2021, 374, 299), treated CTLs performed an enhanced killing capacity to targets (B16F10 cancer cells), especially when the population of CTLs was limited. In a 2-h killing assay (**Fig. 4d**), LDH-modulated CTLs killed ~20% of targets at a CTL-to-target ratio of 0.62:1. By contrast, CTLs from the control group nearly failed to work. It suggested a positive function on CTL activation and cytotoxicity, from either metal cation of LDH adjuvant.

Mg^{2+} ions have been reported to be required by cell-surface molecule lymphocyte function-associated antigen 1 (LFA-1) for an active confirmation on $CD8^+$ T cells, boosting TCR signaling transduction, metabolic programming, and even cytotoxicity (*Cell*, 2022, 185, 585). LFA-1-driven signaling activity including proximal focal adhesion kinase (FAK) and distal extracellular signal-regulated kinase 1/2 (ERK1/2) thus could behave in an Mg^{2+} -modulated manner. As indicated in **Fig. 4e,f**, phosphorylated FAK (pFAK) and ERK1/2 (pERK1/2) were augmented under the stimulation of free Mg^{2+} or Mg^{2+} -containing LDH, but not Fe^{3+} . Blocking the regulation with FAK inhibitor (FAKi; PF-562271) reduced the distal pERK1/2 (**Fig. 4f**), and ultimately CTL activation and killing performance (**Fig. 4a-d**). Mg^{2+} cations from LDH adjuvant could shape CTL immunity successfully (**Fig. 4i**).

Moreover, cellular ROS participates in the modulation of TCR signaling of CTLs via a nuclear factor of activated T-cells (NFAT)-related pathway (*Trends Immunol.*, 2018, 39, 489; *Immunity*, 2013, 38, 225; *Nat. Immunol.*, 2004, 5, 818). To stimulate CTLs with ROS-promoter Fe^{3+} or Fe^{3+} -involving LDH, we identified the translocation of NFAT1 from the cytoplasm to the nucleus (**Fig. 4g**), suggesting that NFAT was activated and performed as a transcription factor. The nuclear import of NFAT1 was not obviously found in Mg^{2+} -treated group, but was downregulated while adding ROS-consumer (glutathione, GSH), which confirmed the ROS-relevant function of Fe^{3+} on NFAT activation. Inhibiting the activation with NFAT inhibitor (NFATi; 11R-VIVIT TFA) interfered the downstream IL-2 induction (**Fig. 4h**) and the contribution of LDH on CTL effector function (**Fig. 4a-d**). Taken together, Fe^{3+} cations from LDH adjuvant programmed CTL immunity through a ROS-NFAT axis (**Fig. 4i**).

Fig. 4 | In vitro activation and cytotoxicity of CTLs regulated by LDH adjuvant. a-c, Representative histograms and quantifications of the expression of early activation marker CD69 (**a**), activation marker CD25 (**b**), and degranulation marker CD107a (**c**) on murine CTLs. **d**, a 2-h killing assay of CTLs, varying BALB/c CTLs to targets (B16F10 cancer cells) ratio. **e,f**, Representative histograms and quantifications of the phosphorylation of FAK (focal adhesion kinase; **e**) and the phosphorylation of ERK1/2 (**f**); **g**, NFAT1 (also known as NFATc2, nuclear factor of activated T cells, cytoplasmic 2) levels in nuclear and cytoplasmic fraction, investigated by western immunoblot. **h**, CTL IL-2 levels in the cellular media. **i**, Representative pathways involved in CTLs immune shaping by LDH adjuvant. For all tests, n = 3 per group. For **a-f** and **h**, data are presented as mean \pm SD, and statistical significance was calculated via one-way ANOVA with Tukey's multiple comparisons. *P < 0.05, **P < 0.01, ***P < 0.001, ****P < 0.0001.

Fig. 3i,j and Fig. 4 have been added into the revised manuscript with the corresponding descriptions in line 29-30, page 12, line 1-26, page 13, and line 13-22, page 15.

8) Please provide the data to confirm the in vivo biosafety of the flex-patch.

Response: To assess the in vivo biosafety of our flex-patch, we harvested the murine heart, liver, spleen, lung, and kidney on postsurgical day 36, and observed no obvious histopathological variations in these tissues (**Supplementary Figure 6a**). To further analyze the biochemical markers in murine sera, the implanted patch barely affected the normal fluctuations of blood urea nitrogen (BUN), creatine kinase (CK), lactate dehydrogenase (LADH), creatinine (CRE), aspartate transaminase (AST), and alanine aminotransferase (ALT) (**Supplementary Figure 6b**). Our flex-patch exhibits good biocompatibility and biosafety for in vivo application.

Supplementary Figure 6 has been added into the revised Supplementary Information. The corresponding descriptions have been replenished into **line 19-27, page 9**, in the revised manuscript, and “**in vitro and in vivo biosafety of patch**” section in the revised Supplementary Information.

Supplementary Figure 6. In vivo biosafety of the flex-patch. a, Hematoxylin and eosin (H&E) staining of major organs harvested from healthy mice and patch-treated mice on postsurgical day 36. Scale bar: 2.5 mm (low-magnification image); scale bar: 100 μm (high-magnification image). **b**, Heatmap exhibiting the relative biochemical levels in murine sera from healthy group and patch-treated group. BUN, blood urea nitrogen; CK, creatine kinase; LADH, lactate dehydrogenase; CRE, creatinine; AST, aspartate transaminase; ALT, alanine aminotransferase. (n = 3 per group)

9) The whole manuscript needs further polishing, including the write logic of the manuscript and the detailed information of the experimental methods.

Response: The whole manuscript has been carefully polished as suggested.

Reviewer #4 (Remarks to the Author): with expertise in cancer immunology, lymphatic immunobiology

The manuscript by Li et al proposes an interesting strategy to treat the postsurgical wound following tumor resection for the purposes of releasing antibodies and adjuvants to the sentinel lymph node for boosted systemic immune surveillance. The approach is innovative and appears to be effective in the 4T1 model used. While the data presented is intriguing, the mechanistic claims are supported only by very descriptive data and not linked functionally to the therapeutic response. It is therefore unclear to what extent the various changes observed are driving outcomes in order to inform optimization or future use of this technology.

Major Comments:

1) LDH is never defined, it is unclear what this adjuvant is and why it was chosen.

Response: A pioneering study reveals that Mg^{2+} ions improve the glycolytic metabolism and cancer-directed effector activity of effector $CD8^+$ T cells via the T cell receptor (TCR) signal strengthening (*Cell*, 2022, 185, 585). Rather than being a metabolic waste that can oxidize cellular components, reactive oxygen species (ROS) have also been reported in numerous studies to show a beneficial role in T cell activation (*Immunity*, 2013, 38, 225; *Trends Immunol.*, 2018, 39, 489). They could enhance TCR signaling via regulating redox-related proteins who participate in T cell activation. Introducing Mg^{2+} and ROS-promoter Fe^{3+} into an immune adjuvant may synergistically influence functional activities of $CD8^+$ T cells in SLN and modulate their immunoresponsive outcomes. As one typical immunoadjuvant nanosheet, layered double hydroxide (LDH) has a sandwich framework composed by edge-sharing $M^{2+}(OH)_6$ and $M^{3+}(OH)_6$ octahedral units ($z = 1$ or 2 ; M, metal cations) on the hydroxide host layer and intercalating exchangeable anions between

the layers (*ACS Nano*, 2021, 15, 2812; *Chem. Soc. Rev.*, 2018, 47, 4954). This unique structure enables LDH a superior adjuvant candidate to integrate functional Mg^{2+} and Fe^{3+} ions for efficient SLN programming.

These contents have been added in the **Introduction** section in the revised manuscript.

2) Figure 2F, it is unclear what is being imaged, assume the inguinal LN basin. Please label the image to facilitate interpretation. An image of an early time point would be more informative, when you expect high doses in the ipsilateral but not contralateral LN. Unclear, why you would still have immunofluorescent tracer at such long time points (5 weeks?) and in distal locations, or what the relevance is for the therapeutic response. Please clarify.

Response: As suggested, we labeled the LN location in revised **Fig. 2i**.

In fact, a quantitative analysis was carried out periodically to present us a comprehensive information on the patch accumulation in LN. As shown in Fig. 2h, no obvious accumulation of the flex-patch in SLN was observed during the initial postoperative two days. Concentration of Fe^{3+} ions rose on 4th day and peaked on 8th day in SLN, which hinted us a SLN imaging around the patch-accumulation maximum (on postoperative 7th day). Here, contralateral LN in distal location was also examined in Fig. 2h, to highlight a more distinct accumulating performance of the flex-patch in SLN than in other LNs.

Moreover, since that a complete disassembling of the implanted flex-patch took at least five weeks in vivo (Fig. 2e and f), we extended quantitative anatomy and found a slight increasing of Fe^{3+} concentration in the contralateral LN on postoperative 36th day. To analyze the experimental results in Fig. 2e-i comprehensively, we think that a sustainable releasing of our flex-patch enabled a five-week immunofluorescent tracer in vivo. Similar phenomena have been reported in other in vivo biodegradable implants (*Nat. Nanotech.* 2021, 16, 538; *Sci. Transl. Med.* 2018, 10, eaar1916). Different from the multiple-injection free agents, a therapeutic implant may trigger/modulate the antitumor immunity with a gentle and long-lasting performance. The corresponding explanation has been added in **line 14-21, page 9**, in the revised manuscript.

Fig. 2 e,f, Fluorescence IVIS imaging (**e**) and quantification (**f**) depicting the sustained retention and disassembling of the implanted patch in vivo. $n = 3$; data are presented as mean \pm SD. **g**, Schematic illustrating experimental details in (**h**) and (**i**). **h,i**, Quantitative analysis (**h**) and intravital imaging (**i**) delineating the SLN accumulation of the patch and its distant migration to the contralateral LN. For (**h**), concentrations of released items were normalized as the percentage of accumulated dose of Fe per gram of LN (%ID g^{-1}). $n = 4$ at each time point per group. Data are presented as mean \pm SD. For (**i**), blood vessel was labeled by FITC-dextran and flex-patch was labeled by Cy5.5, respectively, both of which facilitated an intravital imaging focusing on SLN and its contralateral LN, to track the patch accumulation. Scale bar: 500 μm .

Ultimately, while a slight enrichment of the flex-patch in the contralateral LN to the SLN within postoperative one month (Fig. 2h,i), no any H-2Ld SPSYVYHQF pentamer⁺CD8⁺ T cells was traced (Supplementary Figure 13). As a comparison, of the treated SLNs, H-2Ld SPSYVYHQF pentamer⁺CD8⁺ T cells were distinguished and performed a rapid and robust enhancement from

postsurgical day 5 to day 30, especially in the group V which treated with LDH@aPD-1@CC. (Fig. 5f). The both tests emphasized the importance of metastatic SLN and the high efficacy of our LDH@aPD-1@CC flex-patch on tumor-specific CD8⁺ T activation.

Fig. 5f has been replenished in the revised manuscript with the corresponding context in line 13-17, page 16. In addition, Supplementary Figure 13 has been added in the revised Supplementary Information.

Supplementary Figure 13. Analyses of the LN on the contralateral side of SLN. a, Schematic illustrating the LN survey on 4T1 BC mouse model after surgery, including no flex-patch implantation (untreated, group I) and flex-patch with different therapeutic agents (CC, group II; aPD-1@CC, group III; LDH@CC, group IV; LDH@aPD-1@CC, group V). **b**, Representative flow cytometric plots of H-2Ld SPSYVYHQF-pentamer⁺CD8⁺ T cells, gating on CD8⁺CD3⁺ cells, on postsurgical day 30. n = 5 mice per group.

Fig. 5f. Representative flow cytometric plots of H-2Ld SPSYVYHQF-pentamer⁺CD8⁺ T cells, gating on CD8⁺CD3⁺ cells in SLN, on postsurgical day 5 and day 30. n = 5 mice per group.

3) Please provide controls for the PD-1 immunofluorescence in Figure 2.

Response: To harvest metastatic and healthy SLNs, we identified that 4T1 tumor invasion upregulated cellular PD-1 level in SLN (Supplementary Figure 3), including that of activated CD8⁺

T cells (**Supplementary Figure 4**). According to previous studies, removing tumor-draining sentinel lymph node (SLN) apparently impedes tumor regression induced by the checkpoint administration of PD-1 (*JCI Insight*, 2018, 3, e124507), and decreases intratumoral infiltration of effective tumor-specific CD8⁺ T cells (*Cell*, 2022, 185, 4049). To monitor tumor patients responding to PD-1 blockade therapy positively, phenotypic change of exhausted CD8⁺ T cells in peripheral blood, but not in their intratumoral area, is identified (*Nature*, 2017, 545, 60; *Nat. Med.*, 2019, 25, 1251). It seems that T-effector saving anti-PD-1 antibodies (aPD-1) modulate the protumoral tendency of SLN, and the educated SLN endows aPD-1 improved antitumor effectiveness in return. Therefore, we attempted to load therapeutic aPD-1 into our flex-patch for efficient SLN delivery and programming. PE fluorophore-labeled aPD-1 were well dispersed in the flex-patch, indicated in **Fig. 2a-c**.

Supplementary Figure 3 and Figure 4 have been replenished into the revised Supplementary Information. The corresponding context has been added in **line 24-26, page 3, and line 1-2, page 9**, in the revised manuscript.

Supplementary Figure 3. PD-1 expression in the LN analyzed by IHC staining. PD-1 staining in healthy LN (up) and in metastatic SLN (down). For metastatic SLN, IHC analysis was carried out on post-inoculation day 10 of 4T1 BC mouse model. Scale bar: 100 µm. n = 3 mice per group. Data are presented as mean ± SD.

Supplementary Figure 4. PD-1 expression of murine CTLs stimulated by 4T1 cancer cells. Here, murine CTLs were pre-activated by aCD3/28. Prolonging the co-culture of 4T1 cancer cells and murine CTLs could upregulate the surface PD-1 expression of CTLs. $n = 3$ per group. Data are presented as mean \pm SD.

Fig. 2 a, Representative transmission electron microscopy (TEM) image of LDH nanosheets (scale bar: 50 nm). **b**, Photograph and schematic contents of the flex-patch. **c**, Representative scanning electron microscopy (SEM) image of the flex-patch with elemental mapping (scale bar: 2 μ m). Uniform distribution of aPD-1 labelled with PE fluorophore was imaged by fluorescence microscopy (scale bar: 20 μ m).

4) There are some questions about the number of times the main therapeutic efficacy experiment was performed. Only 6 mice are shown, though the figure legend says the experiment was performed 3 times. Please present all of the data in order to better evaluate the consistency of the therapeutic response. Additionally, therapeutic efficacy is only demonstrated in the 4T1 model, additional models should be employed to demonstrate the broad applicability of this therapy.

Response: For our new in vivo therapeutic trials, five groups including group I (untreated), group II (CC), group III (aPD-1@CC), group IV (LDH@CC), and group V (LDH@aPD-1@CC) have been evaluated on two breast tumor models (4T1 and EMT6, 12 mice per group), to study the broad applicability of our flex-patch.

For 4T1 tumor model, patches loaded with different functional agents were implanted after primary tumor dissection on day 10 of 4T1-GFP-luc inoculation (Fig. 6a). In vivo bioluminescence (Fig. 6b) and tumor volume monitoring (Fig. 6c) showed that mice with surgery only (group I) or with CC matrix (group II) had severe tumor relapse. The aPD-1@CC (group III) and LDH@CC (group IV) patch protected about half of 12 mice from tumor recurrence for at least one month, and nevertheless, was not comparable with the effect of LDH@aPD-1@CC (group V), in which 10 out of 12 mice were tumor-free (Fig. 6b-e). During the treatment, body weights of all mice were not influenced (Fig. 6f). To anatomize all tumor mice and image their main visceral organs, we noticed harsh systemic metastases from group I-IV (Fig. 6g). LDH@aPD-1@CC patch impeded postsurgical 4T1 tumor relapse highly efficiently.

To test the antitumor performance that flex-patch evoked, we rechallenged tumor-free mice collected from group V with 4T1-GFP-luc cells (Fig. 6h). During the following 30 days, distinguishing from the zero survival in control group, all mice after the LDH@aPD-1@CC therapy fought against tumor development successfully (Fig. 6i-k). However, intraperitoneal injection of aCD8 completely subverted the tumor elimination efficacy.

We next established EMT6 BC model to investigate a broad-spectrum application of flex-patch (Supplementary Figure 15a). Surgery and patch implantation were adopted when primary tumor grew up to ~500 mm³ on post-inoculation day 12. To survey the therapeutic effect in each group, LDH@aPD-1@CC patch likewise exhibited best adjuvant performance and inhibited all tumor recurrence (Supplementary Figure 15b-e). In the following rechallenge trial, 87.5% of LDH@aPD-

1@CC-treated mice remained tumor free, which unsurprisingly was invalidated by the aCD8 blocking (Supplementary Figure 15f-i). Together, aPD-1, in concert with LDH immune adjuvant, aroused the postsurgical host an efficient and long-term therapeutic defense against BC relapse, and this defense associates with patch-shaped CTLs.

Fig. 6 and Supplementary Figure 15 have been appended in the revised manuscript and Supplementary Information, respectively. The corresponding context has been added in line 3-27, page 19, in the revised manuscript.

Fig. 6 | Postsurgical 4T1-BC treatment of flex-patch. **a**, Schematic illustrating therapeutic procedure on 4T1 BC mouse model after surgery, including no patch implantation (untreated, group I) and patch with different therapeutic agents (CC, group II; aPD-1@CC, group III; LDH@CC, group IV; LDH@aPD-1@CC, group V). n = 12 mice per group. **b**, Representative bioluminescence images of the mice per group. **c,d**, Individual (**c**) and average (**d**) tumor growth kinetics per group. **e,f**, Mice survival (**e**) and weight changes (**f**) per group. **g**, Representative bioluminescence images of visceral organs of tumor mice in group I-IV. **h**, Schematic illustrating the route of 4T1 tumor rechallenge assay. Apart from the control group, all tumor-free mice were harvested from group V (LDH@aPD-1@CC). n = 8 mice per group. **i,j**, Individual (**i**) and average (**j**) tumor growth kinetics per group. **k**, Mice survival in different rechallenge groups. For (**d,f,j**), curve ended when the first mouse in the corresponding group died. Data are presented as mean \pm SD. Statistical significance was calculated via one-way ANOVA with Tukey's multiple comparisons, ****P < 0.0001. For mice survival analysis (**e,k**), statistical significance was calculated via the Log-rank (Mantel-Cox) test, by comparing with the untreated group I, **P < 0.01, ****P < 0.0001.

Supplementary Figure 15. Postsurgical EMT6-BC treatment of flex-patch. **a**, Schematic illustrating therapeutic procedure on EMT6 BC mouse model after surgery, including no patch implantation (untreated, group I) and patch with different therapeutic agents (CC, group II; aPD-1@CC, group III; LDH@CC, group IV; LDH@aPD-1@CC, group V). $n = 12$ mice per group. **b,c**, Individual (**b**) and average (**c**) tumor growth kinetics per group. **d,e**, Mice survival (**d**) and weight changes (**e**) per group. **f**, Schematic illustrating the route of EMT6 tumor rechallenge assay. Apart from the control group, all tumor-free mice were harvested from group V (LDH@aPD-1@CC). $n = 8$ mice per group. **g,h**, Individual (**g**) and average (**h**) tumor growth kinetics per group. **i**, Mice

survival in different rechallenge groups. For (c,e,h), curve ended when the first mouse in the corresponding group died. Data are presented as mean \pm SD. Statistical significance was calculated via one-way ANOVA with Tukey's multiple comparisons, ****P < 0.0001. For mice survival analysis (d,i), unless indicated otherwise, statistical significance was calculated via the Log-rank (Mantel-Cox) test, by comparing with the untreated group I, ***P < 0.001, ****P < 0.0001.

5) It is surprising that 2/5 tumors grew out in the re-challenged mice. A larger n would be helpful here. Can the authors provide data to explain the heterogeneity in this re-challenge experiment?

Response: Inoculating two millions of 4T1 cancer cells caused 3/5 of rechallenging success. However, to reduce the inoculated cells by half in the new 4T1 rechallenge trial, we observed that 8/8 of mice fought against tumor growth successfully (n = 8, LDH@aPD-1@CC, Fig. 6i). By contrast, intraperitoneal injection of aCD8 subverted the above antitumor result (n = 8, LDH@aPD-1@CC + aCD8, Fig. 6i).

To conduct the same rechallenging trial in EMT6 tumor model, 7/8 of LDH@aPD-1@CC-treated mice maintained tumor free (n = 8, LDH@aPD-1@CC, Supplementary Figure 15g), which unsurprisingly was invalidated by the aCD8 blocking (n = 8, LDH@aPD-1@CC + aCD8, Supplementary Figure 15g). These rechallenging differences may associate with the number ratio of patch-shaped cytotoxic T lymphocytes (CTLs) to the inoculated cancer cells, supported by some studies where CTLs perform tumoricidal effect in a dose-dependent manner (*Nat. Commun.*, 2021, 12, 3229; *Science*, 2021, 374, 299).

Fig. 6i and Supplementary Figure 15g have been appended in the revised manuscript and Supplementary Information, respectively. The corresponding context has been added in line 14-18 and line 23-27, page 19, in the revised manuscript.

Fig. 6 i, Individual 4T1 tumor growth kinetics per group in rechallenge assay. n = 8 mice per group.

Supplementary Figure 15 g, Individual EMT6 tumor growth kinetics per group in rechallenge assay. n = 8 mice per group.

6) In discussion of Figure 3, the language used in the text makes it very difficult to follow, please consider revising. For example, was it one mouse in group IV that showed no metastasis? Only the one mouse that had a local recurrence was evaluated?

Response: As suggested, in vivo therapeutic tests have been modified with five groups: group I (untreated), group II (CC), group III (aPD-1@CC), group IV (LDH@CC), and group V (LDH@aPD-1@CC), on two breast tumor models (4T1 and EMT6, 12 mice per group), to study the broad applicability of our flex-patch. The corresponding discuss has been carefully polished in line 3-27, page 19, in the revised manuscript.

7) The latency of the growth of the EMT6 tumors is 150 days? Can the authors provide growth rates in naive mice? Is this an impact of the treatment or do these tumors simply.

exhibit such poor growth. It is hard to interpret this experiment and accept the claims of the authors when the 4T1 outgrowth in re-challenged mice is so much faster than the antigen-independent model.

Response: We apologize for the misunderstanding we caused. In fact, as shown in the Previous Fig. 3, to investigate whether the patch-triggered antitumor immunity exhibits tumor specificity, EMT6 challenging (Previous Fig. 3j) was carried out by following an earlier 4T1 challenging (Previous Fig. 3f). That is why the actual inoculation time for EMT6 cancer cells were on 130th day if counting from the in vivo test began. Moreover, the EMT6 growth rates in naïve mice had been shown in Previous Supplementary Figure 5d.

Previous Fig. 3 a,f,j, Schematic illustrating therapeutic procedure on 4T1 BC mouse model after surgery.

Previous Supplementary Figure 5. d, Average tumor growth kinetics in different groups. Here the EMT6 control indicates the inoculation of EMT6 on naïve mice.

Now, as we explained in response #4 and #6, in vivo therapeutic tests have been modified with five groups: group I (untreated), group II (CC), group III (aPD-1@CC), group IV (LDH@CC), and group V (LDH@aPD-1@CC), on two breast tumor models (4T1 and EMT6, 12 mice per group), to study the broad applicability of our flex-patch. The corresponding **Fig. 6** (for 4T1 tumor model) and **Supplementary Figure 15** (for EMT6 tumor model) have been well organized in the revised manuscript with the discussion in **line 3-27, page 19**.

8) The authors make a big point of using Cell tag technology to demonstrate that there is a viable metastasis in the LN at time of resection and treatment, but direct imaging of the LN would be more informative. It is not clear what the overlap in bar codes means in terms of metastatic size or how consistent this is across animals. Please expand and clarify as this is made to be an important assumption in their model.

Response: The number of barcodes identified in SLN (2827 in total) indicates how many 4T1 cancer cells could migrate out from the primary site and colonize in this lymphoid tissue (**Supplementary Figure 2d,e**). The barcode overlapping tells us that 2144 barcodes in SLN were also detected in primary tumor, reconfirming the origin of invaded 4T1 cancer cells in SLN.

As reviewer suggested, we examined the SLN metastasis by labeling 4T1 cancer cells with GFP and luciferase (4T1-GFP-luc). On day 10 post inoculation of 4T1-GFP-luc cells, we collected brachial LNs, axillary LNs, and inguinal LNs from the mouse model and identified the luminescent SLN (**Supplementary Figure 2a,b**), which resulted from an invasion of neighboring cancer cells. Meanwhile, these infiltrated cancer cells could also be recognized by detecting their strong GFP expression using immunohistochemistry (IHC) analysis (**Supplementary Figure 2c**).

Supplementary Figure 2 has been updated in the revised Supporting Information. In addition, the corresponding context has been added in **line 26-30, page 8, line 5-9, page 9**, and in Methods of **“Identifying tumor metastasis in SLN”** section in the revised manuscript.

Supplementary Figure 2. SLN metastatic tracking in postsurgical 4T1 BC mouse model. a, Schematic illustrating the inoculation of 4T1-GFP-luc cancer cells to survey SLN metastasis. **b,** Ex vivo bioluminescence of LNs collected from 4T1-tumor model on post-inoculation day 10. Here, LNs include two brachial LNs, two axillary LNs, and two inguinal LNs. The one next to the 4T1 primary tumor is regarded as SLN, which exhibited bioluminescence. $n = 6$ mice. **c,** IHC staining for GFP-labeled 4T1 cancer cells infiltrated in SLN. Scale bar: 100 μ m. $n = 6$ mice. **d,** Schematic illustrating 4T1-celltagging trial to survey SLN metastasis. **e,** 4T1-barcode numbers in SLN on day 10 post inoculation of 4T1-tag cells, quantified by DNA sequencing. $n = 10$ mice.

9) Please provide more rationale for the use of the pentamer described in the text and antigen target. It is presumed from the reference this is the immunodominant AH1 epitope. Please confirm. The paper referenced doesn't actually track this response in 4T1 and in fact shows cross-reactivity with CT26 that appears to be in part independent of AH1. Please provide additional references where this pentamer is used to track antigen-specific responses in 4T1 tumors and also provide controls for the pentamer stain (if already in the suppl fig please

label more carefully). Please also show the pentamer stain with CD44 to ensure an antigen-experienced phenotype.

Response: In fact, H-2Ld-restricted bioactive nanomeric peptide, AH1, with the sequence SPSYVYHQF, is derived from envelope glycoprotein 70 (gp70) of endogenous murine leukemia virus, and is expressed in several murine tumors including 4T1. Specifically, relative expression level of gp70-coding transcripts in 4T1 cancer cell line is as high as that in CT26 (*Oncoimmunology*, 2013, 2, e26889). Although the paper referenced does not show the experimental result of AH1-specific response in 4T1 tumor, it exhibits the feasibility in the introduction section. Here, additional references have been provided to confirm that AH1 works as the immunodominant antigen of 4T1 tumor (*Oncoimmunology*, 2013, 2, e26889; *Cancer Immunol. Immunother.*, 2021, 70, 3183; *Cancer Immunol. Immunother.*, 2020, 69, 2063; *Cancer Immunol. Immunother.*, 2003, 52, 739). In references of *Cancer Immunol. Immunother.*, 2021, 70, 3183 and *Blood*, 2003, 101, 1645, AH1-tetramers were applied to track AH1-specific CD8⁺ T cells in 4T1 tumor model successfully. In our test, AH1-pentamers were selected for their stability and for the increased intensity and specificity of the positive population compared to the corresponding tetramers (www.proimmune.com).

Furthermore, pentamer loaded with H-2Ld MVPGGQSSF peptide, an immunodominant epitope originating from mycobacterium tuberculosis, has been customized as control. To stain CD8⁺ T cells collected from SLN with these pentamers, no H-2Ld MVPGGQSSF⁺ subset was discovered (Supplementary Figure 12c).

Supplementary Figure 12. c, Representative flow cytometric plots of H-2Ld MPVGGQSSF-pentamer⁺CD8⁺ T cells, gating on CD8⁺CD3⁺ cells. n = 5 mice per group. SLNs in each group were harvested on postsurgical day 5 and day 30, respectively.

To quantify the phenotypic variation of H-2Ld SPSYVYHQF⁺CD8⁺ T cells during the patch treatment, we stained these cells with CD44 and CD62L. On postsurgical day 5, H-2Ld SPSYVYHQF⁺CD8⁺ T cells exhibited high percentage of effector memory (T_{em}; CD62L⁻CD44⁺) cells (Fig. 5j). To analyze again on postsurgical day 30, this subset decreased significantly and central memory (T_{cm}; CD62L⁺CD44⁺) cells dominated. According to previous studies, T_{em} cells display immediate effector function in response to antitumor therapy, whereas T_{cm} cells mediate reactive memory, proliferating and differentiating to effector T if antigen stimulates again (*Nat. Commun.* 2017, 8, 1954; *Annu. Rev. Immunol.* 2004, 22, 745). LDH@aPD-1@CC empowers retained SLN a long-term tumor immunity against cancer resurgence.

Fig. 5j and Supplementary Figure 12c have been replenished in the revised manuscript and Supplementary Information, respectively. The corresponding descriptions have been added in line 6-17 and line 24-26, page 16 in the revised manuscript.

Fig. 5 j, Representative flow cytometric plots of CD8⁺ T_{em} and CD8⁺ T_{cm} cells, gating on H-2Ld SPSYVYHQF-pentamer⁺CD8⁺CD3⁺ cells in SLN.

10) While the patch clearly has an effect on tumor recurrence in their 4T1 model, the mechanistic insight into why this is, is very limited. Markers used to define the single cell populations are very limited and the data is only descriptive, there was no functional validation of the proposed metabolic changes observed in either the CD4 or CD8 compartment. More importantly, however, it is not clear that any of these metabolic changes would even be causal as it is already known that activating T cells will remodel their metabolic state to permit rapid proliferation and effector function. Therefore, whether the differences observed really represent a direct effect of the therapy that then drives more efficient effector function vs a correlate with the activation that is ongoing is unclear. The comparisons made are also only between the combination therapy and unloaded patch, what is the metabolic effect of the individually loaded LDH and PD-1 patches? The Treg data is interesting but with no functional followup it is unclear how this contributes to the therapeutic phenotype observed.

Response: We totally agree that the mechanistic insight is very limited in the initial manuscript, which has been comprehensively explored in our revision with functional validation.

First, we revealed that aPD-1 and LDH modulate the metabolic activities of activated CD8⁺ T cells in SLN.

From the collected SLNs, we identified the activated CD8⁺ T cells with scRNA seq (Fig. 3a,b), to investigate their differentially expressed genes (DEGs). Distinct from CC treatment, activated CD8⁺ T cells after aPD-1@CC therapy (Fig. 3c) exhibited the up-regulation of TCR signaling (*VPS37B*), effector function (*IFRD1*), and survival (*HSPA5*, *MAP1LC3B*) (*Gene*, 2017, 618, 14; *Nat. Immunol.*, 2014, 15, 1152). Similar improvements involving TCR signaling (*VPS37B*), activation (*AMD1*, *IRS2*), effector function (*IFRD1*), proliferation (*DUSP10*), and persistence (*FOSL2*), were also pronounced (Fig. 3d) after LDH@CC management (*Immunity*, 2011, 35, 871; *J. Biol. Chem.*, 1995, 270, 28527; *Nature*, 2004, 430, 793; *Cell Rep.*, 2018, 23, 2142). Gene set enrichment analysis (GSEA) on DEGs supported these variations. In immunosuppressive SLN (CC, Fig. 3e,f), mitochondrial activities of activated CD8⁺ T cells featured three important pathways of citric acid cycle (also known as TCA cycle), respiratory electron transport, and oxidative phosphorylation (also known as OXPHOS). But none of these pathways enriched after aPD-1@CC or LDH@CC administration. According to pioneering studies, TCA cycle and OXPHOS pathways indicate cellular quiescence with basal metabolic demand and represent in naïve and memory T cells generally, but not proliferating effector T cells (*Science*, 2013, 342, 1242454; *Nat. Rev. Immunol.*, 2005, 5, 844). SLN immunosuppression seems turned activated CD8⁺ T cells to be a resting-like state.

To confirm the metabolic activity modulated by aPD-1 and LDH adjuvant, we co-cultured these functional agents with murine cytotoxic T lymphocytes (CTLs) in vitro. These CTLs were pre-activated by anti-CD3 and anti-CD28 antibodies (aCD3/28). To further stimulate them with aCD3/28, we observed a transient rising of glycolysis followed by sharp dropping (Fig. 3g). Replenishing aPD-1, Mg²⁺ or Fe³⁺ (alkaline LDH cannot be used directly since it severely affect the extracellular acidification rate (ECAR) measured in this test) intriguingly maintained the glycolytic activity in diverse degrees (Fig. 3h). This up-regulated glycolysis has been revealed to switch quiescent T cells to highly effective states (*Science*, 2013, 342, 1242454), which explains the DEGs signature (Fig. 3c,d) in activated CD8⁺ T cells after the engagement of aPD-1 and LDH.

Fig. 3 | Regulated metabolic activities of activated CD8⁺ T cells in SLN after aPD-1 and LDH treatment. **a**, Schematic illustrating the scRNA seq trial on 4T1 BC mouse model after surgery, including no flex-patch implantation (untreated) and flex-patch with different therapeutic agents (CC, aPD-1@CC, and LDH@CC). **b**, UMAP plot highlighting the identified activated CD8⁺ T cells. **c,d**, Volcano plots showing DEGs in activated CD8⁺ T cells between aPD-1@CC versus (vs.) CC (**c**) and LDH@CC vs. CC (**d**). Significant DEGs (blue and red dots): p value adjusted < 0.05 and fold change > 2. **e,f**, Representative GSEA-enriched pathways for activated CD8⁺ T cells in aPD-1@CC vs. CC (**e**) and LDH@CC vs. CC (**f**). **g,h**, Glycolytic activity of murine CTLs with diverse treatments. Glycolytic activity (Δ ECAR) is determined by subtracting ECAR maximum from baseline. n = 3 per group. Data are presented as mean \pm SD. Statistical significance was calculated via one-way ANOVA with Tukey's multiple comparisons. *P < 0.05, **P < 0.01, ***P < 0.001. **i**, Representative GSEA-enriched pathways for activated CD8⁺ T cells, in LDH@CC vs. CC. GSEA p value represents the familywise-error rate (FWER) p value; NES, normalized enrichment score; FDR, false discovery rate. n = 6 per group. Significantly enriched pathways with GSEA FWER p value < 0.05. **j**, RT-qPCR validation of scRNA seq data for enriched top 10-ranked genes from (**i**) in murine CTLs. Relative quantitation was obtained by normalization to 18S expression. n = 3 per group. Data are presented as mean \pm SD. Statistical significance was calculated via paired two-tailed t-test. *P < 0.05, **P < 0.01, ***P < 0.001, and ****P < 0.0001.

To continue the GSEA on DEGs, we identified an obvious contribution of LDH adjuvant on the initiation of eukaryotic translation (**Fig. 3i**). To validate the expression of top 10-ranked genes from this enrichment by quantitative real-time polymerase chain reaction (RT-qPCR), increased transcription of EIF5, RPL5, RPS6, RPL7, RPS15, RPS14, RPL17, RPL15, RPL13A, and RPL18A was found in LDH@CC-treated CTLs, compared to CC group (**Fig. 3j**). Our data indicated a remarkable influence of LDH adjuvant to CTL biology.

Second, we demonstrated that LDH adjuvant promotes the activation and cytotoxicity of cytotoxic T lymphocytes (CTLs) via metal cation-mediated programming.

Two functional metal cations (Mg²⁺ and Fe³⁺ ions) from the LDH adjuvant were separately analyzed here. As exhibited in **Fig. 4a-c**, in the aCD3/28-activating process (aCD3/28, anti-CD3 and anti-CD28 antibodies), either Mg²⁺ or Fe³⁺ enhanced the expression of CD69 (early activation

marker), CD25 (activation marker), and CD107a (degranulation marker) of CTLs. To utilize the MHC mismatch between BALB/c CTLs and B16F10 cancer cells for a cytotoxic test (*Nat. Commun.*, 2017, 8, 511; *Science*, 2021, 374, 299), treated CTLs performed an enhanced killing capacity to targets (B16F10 cancer cells), especially when the population of CTLs was limited. In a 2-h killing assay (**Fig. 4d**), LDH-modulated CTLs killed ~20% of targets at a CTL-to-target ratio of 0.62:1. By contrast, CTLs from the control group nearly failed to work. It suggested a positive function on CTL activation and cytotoxicity, from either metal cation of LDH adjuvant.

Mg²⁺ ions have been reported to be required by cell-surface molecule lymphocyte function-associated antigen 1 (LFA-1) for an active confirmation on CD8⁺ T cells, boosting TCR signaling transduction, metabolic programming, and even cytotoxicity (*Cell*, 2022, 185, 585). LFA-1-driven signaling activity including proximal focal adhesion kinase (FAK) and distal extracellular signal-regulated kinase 1/2 (ERK1/2) thus could behave in an Mg²⁺-modulated manner. As indicated in **Fig. 4e,f**, phosphorylated FAK (pFAK) and ERK1/2 (pERK1/2) were augmented under the stimulation of free Mg²⁺ or Mg²⁺-containing LDH, but not Fe³⁺. Blocking the regulation with FAK inhibitor (FAKi; PF-562271) reduced the distal pERK1/2 (**Fig. 4f**), and ultimately CTL activation and killing performance (**Fig. 4a-d**). Mg²⁺ cations from LDH adjuvant could shape CTL immunity successfully (**Fig. 4i**).

Moreover, cellular ROS participates in the modulation of TCR signaling of CTLs via a nuclear factor of activated T-cells (NFAT)-related pathway (*Trends Immunol.*, 2018, 39, 489; *Immunity*, 2013, 38, 225; *Nat. Immunol.*, 2004, 5, 818). To stimulate CTLs with ROS-promoter Fe³⁺ or Fe³⁺-involving LDH, we identified the translocation of NFAT1 from the cytoplasm to the nucleus (**Fig. 4g**), suggesting that NFAT was activated and performed as a transcription factor. The nuclear import of NFAT1 was not obviously found in Mg²⁺-treated group, but was downregulated while adding ROS-consumer (glutathione, GSH), which confirms the ROS-relevant function of Fe³⁺ on NFAT activation. Inhibiting the activation with NFAT inhibitor (NFATi; 11R-VIVIT TFA) interfered the downstream IL-2 induction (**Fig. 4h**) and the contribution of LDH on CTL effector function (**Fig. 4a-d**). Taken together, Fe³⁺ cations from LDH adjuvant program CTL immunity through a ROS-NFAT axis (**Fig. 4i**).

Fig. 4 | In vitro activation and cytotoxicity of CTLs regulated by LDH adjuvant. a-c, Representative histograms and quantifications of the expression of early activation marker CD69 (**a**), activation marker CD25 (**b**), and degranulation marker CD107a (**c**) on murine CTLs. **d**, a 2-h killing assay of CTLs, varying BALB/c CTLs to targets (B16F10 cancer cells) ratio. **e,f**, Representative histograms and quantifications of the phosphorylation of FAK (focal adhesion kinase; **e**) and the phosphorylation of ERK1/2 (**f**); **g**, NFAT1 (also known as NFATc2, nuclear factor of activated T cells, cytoplasmic 2) levels in nuclear and cytoplasmic fraction, investigated by western immunoblot. **h**, CTL IL-2 levels in the cellular media. **i**, Representative pathways involved in CTLs immune shaping by LDH adjuvant. For all tests, n = 3 per group. For **a-f** and **h**, data are presented as mean \pm SD, and statistical significance was calculated via one-way ANOVA with Tukey's multiple comparisons. *P < 0.05, **P < 0.01, ***P < 0.001, ****P < 0.0001.

In summary,

i) The scRNA seq identified that activated CD8⁺ T cells (CTLs) from metastatic SLN enrich citric acid cycle and oxidative phosphorylation pathways, which cannot support their bioenergetic requirements for killing. Delivered aPD-1 and LDH save immunosuppressive CTLs with upregulated glycolytic activity, validated by the glycolysis stress tests on murine CTLs.

ii) LDH adjuvant, beyond our expectation, promotes CTL activation and cytotoxic killing via the metal cation-mediated programming, which has been functionally demonstrated by co-culturing LDH, murine CTLs, and optional inhibitors through a series of in vitro tests.

Finally, in the LDH@aPD-1@CC patch-driven SLN, tumor-specific CD8⁺ T memory cells generate and differentiate obviously, protecting postsurgical mice from high-incident BC recurrence and rechallenge.

Abstract and **Introduction** have been revised; **Fig. 3** and **Fig. 4** have been added in the revised manuscript with the corresponding descriptions in **line 11-25, page 10, line 20-30, page 12, line 1-26, page 13, and line 13-22, page 15.**

11) The authors comment on the immune cells that are recruited into the surgical site with and without the patch but make no reference to the rate of healing in these various groups. Can they please provide data on wound closure? Also, while the biology dissected in Fig 7 is likely relevant to the responses observed it is again never functionally linked to the therapeutic response and unclear how or if it contributes.

Response: Postsurgical macrophages re-educated by aPD-1 and LDH exhibited predominant M1 phenotype (high CD80⁺CD86⁺ and low CD206⁺ levels, group V in Fig. 7c-h), which may recruit more CD8⁺ T cells (*J. Clin. Invest.*, 2019, 129, 5151; *J. Immunother. Cancer*, 2020, 8, e000778). Here, in contrast to the untreated (group I, Fig. 7i,k), much more CD8⁺ T cells infiltrated in patch-treated postsurgical tumor residual (group V, Fig. 7i,k). This high infiltration disappeared while inhibiting macrophages with CSF-1R antibodies (aCSF-1R; Fig. 7k). Ultimately, recruited CD8⁺ T cells eliminated tumor residual efficiently, supported by a high level of cleaved caspase-3 (CC3, apoptosis marker) from group V, Fig. 7l.

For these patch-treated mice, although much higher levels of pro-inflammatory TNF- α , IL-12, and interferon γ (IFN- γ) cytokines and decreased anti-inflammatory IL-10 in sera on postoperative day 5 (Supplementary Figure 22a,b), all of them rapidly re-modulated during the following 10 days (Supplementary Figure 22c). It may explain an unaffected wound healing without tumor relapse in group V, compared to group I (Fig. 7m).

Fig. 7 and Supplementary Figure 22 have been added in the revised manuscript and Supplementary Information, respectively. The corresponding contents have been prepared in line 15-25, page 22, in the revised manuscript.

Fig. 7 | Immune actions of flex-patch to postsurgical wound. Tumor residual per mouse was analyzed on postsurgical day 5. Wound healing per mouse was monitored in postsurgical day 1-

15. **a**, Schematic illustrating the tumor residual survey and wound healing on 4T1 BC mouse model after surgery, including no patch implantation (untreated, group I) and patch with different therapeutic agents (CC, group II; aPD-1@CC, group III; LDH@CC, group IV; LDH@aPD-1@CC, group V). **b**, Proportions of CD45⁺ cells and CD45⁺ subsets infiltrated in postsurgical TME (n = 5 mice per group). **c-h**, Representative flow cytometric plots (**c,d,e**) of PD-1⁺TAMs, CD206⁺TAMs, CD80⁺CD86⁺ TAMs and their flow cytometric quantifications (**f,g,h**), gating on CD11b⁺F4/80⁺CD45⁺ cells (n = 5 mice per group). **i,j**, Flow cytometric quantification of CD8⁺ T (**i**) and CD4⁺ T cells (**j**), gating on CD3⁺CD45⁺ cells (n = 5 mice per group). **k,l**, IHC detection and quantification of infiltrated CD3⁺ T and CD8⁺ T cells (**k**), and 4T1 tumoral proliferation and apoptosis (**l**; GFP, 4T1 marker; Ki67, cell proliferation marker; CC3, cell apoptosis marker) in postsurgical tumor residual. Scale bar: 100 μ m; n = 3 mice per group. Data are presented as mean \pm SD. **m**, Photographs of postsurgical wound in group I (untreated) and group V (LDH@aPD-1@CC). n = 3 mice per group. For **f, g, h, i, j**, statistical significance was calculated via one-way ANOVA with Tukey's multiple comparisons. *P < 0.05, **P < 0.01, ****P < 0.0001.

Supplementary Figure 22. Cytokine levels in sera. **a**, The route of blood sera assay. **b,c**, Hematic TNF- α , IL-12, IFN- γ , and IL-10 levels on postsurgical day 5 (**b**) and day 15 (**c**). $n = 5$ mice per group; statistical significance was calculated via one-way ANOVA with Tukey's multiple comparisons. * $P < 0.05$, ** $P < 0.01$, *** $P < 0.001$, **** $P < 0.0001$.

12) Throughout the language used is very difficult to interpret, please revise and simplify the language. Words like anatomize, for example, are used to explain simpler concepts of determining metastatic burden, for example. Line 284-287, is another example of text that is difficult to follow.

Response: We apologize for the obscure language we used. The whole manuscript has been carefully revised and polished.

REVIEWER COMMENTS

Reviewer #1 (Remarks to the Author):

The authors have addressed most of my concerns and the newly supplemental data really strengthened the conclusion. One more minor suggestion, consistent with the luciferase luminescence detection in Supplementary Fig. 2b, GFP expression detected by IHC staining should also include representative image of the control group.

Reviewer #2 (Remarks to the Author):

The authors have addressed most of my concerns. My major suggestion in this round of peer review is that the presentation needs significant improvement to make the manuscript more informative and readable for the audience.

Several detailed comments:

1. Supplementary Fig. 8c shows that the proportion of T cells decreased in SLN implanted with aPD-1@CC and LDH@CC compared with CC group. Which subtype of T cells dominates such changes in the T cell population? Besides, the results of the flow cytometric assay show that the number of CD8+ T increased in aPD-1@CC and LDH@CC groups (Fig. 5b and e). It is necessary to check whether the single-cell data shows the same trends.
2. Some sentences or phrases are repetitive. For example, in the legend of figure 3, "Data are presented as mean \pm SD"; "**P < 0.05, **P < 0.01, ***P < 0.001".
3. Fig3a, the schematic illustration of the scRNA seq experiment is misleading and less informative. Inconsistent with the statement in the main text, the current figure shows that only one tumor sample was collected for sequencing experiments. Besides, this figure should illustrate how scRNA seq experiments were carried out and how many cells were harvested for scRNA seq analysis.
4. Title of the Y-axis in Fig3 c and d, "p value adjustedj" should be "adjusted p value".
5. Bad grammar and confused statements. e.g. "To further stimulate them with aCD3/28, we observed a transient rising of glycolysis followed by sharp dropping"; "To continue the GSEA on DEGs, we identified an obvious contribution of LDH adjuvant on the initiation of eukaryotic translation (Fig. 3i), but not aPD-1 (Supplementary Figure 10)".

Reviewer #3 (Remarks to the Author):

The authors have addressed all the issues.

Reviewer #4 (Remarks to the Author):

The authors present a significantly revised manuscript that has addressed most of the concerns raised on initial review. New data further supports the claim that LDH is an adjuvant meant to deliver Mg²⁺ and Fe³⁺ to promote T cell glycolysis. There remain some issues to be addressed.

2. While slow release is potentially an important feature, it does not explain uptake in the contralateral node and raises potential concern for technical issues with the imaging. Further, quantitative analysis of the Fe seems more reliable and shows relatively little to no accumulation in the contralateral node. The images in 2i are concerning and distract from the point already made in 2h, I would recommend their removal.

4. Thank you for adding the second model.

Figure 3i, I would consider a rechallenge success rate to be tumor growth not failure to grow. If you want to show ratios of 8/8 I would say rate of protection.

5. Thank you.

6. I cannot find where this comment was addressed.

8. I would recommend removing the cell tagging now that direct IHC has been included.

9. New Figure 5, there is a significant number of CD44-CD62L+ naïve pentamer+ cells (70%). This should not be, as the precursor frequency of endogenous T cells is too low for detection without significant enrichment. The authors should only quantify the antigen-specific response by quantifying CD44+tetramer+ cells (both Tem and Tcm). The authors also still need to provide a control for their specific pentamer, while the irrelevant pentamer is a good control it doesn't control of issues with the specificity of the AH1 pentamer in your model. Please show staining in a non-tumor bearing mouse.

10. The added mechanistic data is all very interesting but a bit disconnected. I would recommend removing 3i and j as there is not link to any of the other data. Does anything in Figure 4 relate to the glycolysis findings in Figure 3? Please provide statistics for Figure 4d or remove claims of improved tumor cell killing at limiting dilutions. It is recommended that the key mechanistic insight be highlighted and the rest removed to focus the message.

11. Thank you for the images, but wound closure data should be quantified across all groups. This data is important with respect to the likelihood this kind of approach would ever be taken in patients. The rate of local recurrence should also be quantified. If no difference in rate of local recurrence I would recommend removing the immunophenotyping data at the wound site as it does not add in a meaningful way to explaining the therapeutic response driven by the patch.

As mentioned in the first review, the language remains overly complicated and should be significantly revised and simplified. It is very difficult to extract the key points and pieces of data from the text and figures.

REVIEWER COMMENTS

Reviewer #1 (Remarks to the Author):

The authors have addressed most of my concerns and the newly supplemental data really strengthened the conclusion. One more minor suggestion, consistent with the luciferase luminescence detection in Supplementary Fig. 2b, GFP expression detected by IHC staining should also include representative image of the control group.

Response: Thank you so much for the positive comment. As suggested, the representative image of GFP staining for healthy LN (Supplementary Figure S2c) where no visible GFP signal were observed has been prepared in the revised Supplementary Information.

Supplementary Figure 2. SLN metastatic tracking in postsurgical 4T1 BC mouse model. a, Schematic illustrating the inoculation of 4T1-GFP-luc cancer cells to survey SLN metastasis. **b,** Ex vivo bioluminescence of LNs collected from 4T1-tumor model on post-inoculation day 10. Here, LNs include two brachial LNs, two axillary LNs, and two inguinal LNs. The one next to the 4T1 primary tumor is regarded as SLN, which exhibited bioluminescence. **c, IHC staining of GFP (4T1 marker) in healthy LN (up) and in metastatic SLN (down).** Scale bar: 100 μm. n = 3 mice per group.

Reviewer #2 (Remarks to the Author):

The authors have addressed most of my concerns. My major suggestion in this round of peer review is that the presentation needs significant improvement to make the manuscript more informative and readable for the audience.

Several detailed comments:

1. Supplementary Fig. 8c shows that the proportion of T cells decreased in SLN implanted with aPD-1@CC and LDH@CC compared with CC group. Which subtype of T cells dominates such changes in the T cell population? Besides, the results of the flow cytometric assay show that the number of CD8⁺ T increased in aPD-1@CC and LDH@CC groups (Fig. 5b and e). It is necessary to check whether the single-cell data shows the same trends.

Response: In flow cytometric analysis, what we quantified was not the number of CD8⁺ T cells, but the proportions of CD8⁺ T cells with high proliferation (Ki67⁺), strong killing performance (GzmB⁺), and tumor specificity (H-2Ld SPSYVYHQF pentamer⁺) occupied on total CD8⁺ T cells. For these immune lymphocytes, the treatment of aPD-1@CC or LDH@CC improved their antitumor effector function.

In scRNA seq trial, the proportion of T cells was calculated on all identified cells in SLN, suggesting that other types of cells varied by the therapy (e.g., upregulated B-cell ratio) would influence the T-cell percentage. To analyze CD8⁺ T cells, activated part was increased by LDH adjuvant and kept steady after aPD-1 treatment (see the following **Supplementary Figure 9b**), which kept consistent with our finding that LDH adjuvant promotes CD8⁺ T cell activation, unlike aPD-1 that only works on activated T cells (by blocking the PD-1 presented on the surface (*Sci. Adv.*, 2020, 6, eabd2712; *Immunity*, 2022, 55, 512)).

Considering that the key point in scRNA seq is to study the metabolic reprogramming of activated CD8⁺ T cells by patch therapeutic variables (aPD-1 or LDH adjuvant), we replaced the previous Supplementary Figure 8c with the new **Supplementary Figure 9b** in the revised Supplementary Information.

Supplementary Figure 9b, The proportion of activated CD8⁺ cells in total CD8⁺ cells per group.

2. Some sentences or phrases are repetitive. For example, in the legend of figure 3, “Data are presented as mean ± SD”; “*P < 0.05, **P < 0.01, *P < 0.001”.**

Response: Thanks for the suggestion. “*P < 0.05, **P < 0.01, ***P < 0.001, and ****P < 0.0001” in each figure legend has been moved into the “Statistical analysis” section of the Methods. Phrase of “Data are presented as the mean ± SD” has been simplified as well.

3. Fig3a, the schematic illustration of the scRNA seq experiment is misleading and less informative. Inconsistent with the statement in the main text, the current figure shows that only one tumor sample was collected for sequencing experiments. Besides, this figure should illustrate how scRNA seq experiments were carried out and how many cells were harvested for scRNA seq analysis.

Response: Thanks for the suggestion. **Fig. 3a** has been modified in the revised manuscript.

Fig. 3a, Schematic illustrating the scRNA seq trial on the 4T1 BC mouse model after surgery, including no patch implantation (untreated) and patches with different therapeutic agents (CC, aPD-1@CC, LDH@CC, and LDH@aPD-1@CC).

4. Title of the Y-axis in Fig3 c and d, “p value adjustedj” should be “adjusted p value”.

Response: “p value adjusted” has been corrected to “adjusted p value” (Fig. 3c and d) in the revised manuscript.

5. Bad grammar and confused statements. e.g. “To further stimulate them with aCD3/28, we observed a transient rising of glycolysis followed by sharp dropping”; “To continue the GSEA on DEGs, we identified an obvious contribution of LDH adjuvant on the initiation of eukaryotic translation (Fig. 3i), but not aPD-1 (Supplementary Figure 10)”.

Response: By following the recommendation from the editor, the whole manuscript has been polished by the highly qualified native English speaking editors from the “American Journal Experts”.

Reviewer #4 (Remarks to the Author):

The authors present a significantly revised manuscript that has addressed most of the concerns raised on initial review. New data further supports the claim that LDH is an adjuvant meant to deliver Mg²⁺ and Fe³⁺ to promote T cell glycolysis. There remain some issues to be addressed.

2. While slow release is potentially an important feature, it does not explain uptake in the contralateral node and raises potential concern for technical issues with the imaging. Further, quantitative analysis of the Fe seems more reliable and shows relatively little to no accumulation in the contralateral node. The images in 2i are concerning and distract from the point already made in 2h, I would recommend their removal.

Response: Thank you so much for the comprehensive consideration. As suggested, we have removed Fig. 2i and the corresponding information in the revised manuscript.

4. Thank you for adding the second model.

Figure 3i, I would consider a rechallenge success rate to be tumor growth not failure to grow. If you want to show ratios of 8/8 I would say rate of protection.

Response: Thank you for the advice. In Fig. 7b and Supplementary Figure 19g, “rechallenge success rate” has been modified as “rate of protection”. Moreover, “recurrence rate” has been changed to “rate of recurrence” (Fig. 6c and Supplementary Figure 19b) for same format.

5. Thank you.

Response: It is my honor.

6. I cannot find where this comment was addressed.

Response: By following the recommendation from the editor, the whole manuscript has been polished by the highly qualified native English speaking editors from the “American Journal Experts”.

8. I would recommend removing the cell tagging now that direct IHC has been included.

Response: Thank you so much for the suggestion. All contents on 4T1 cell tagging have been removed in the revised manuscript.

9. New Figure 5, there is a significant number of CD44-CD62L+ naïve pentamer+ cells (70%). This should not be, as the precursor frequency of endogenous T cells is too low for detection without significant enrichment. The authors should only quantify the antigen-specific response by quantifying CD44+tetramer+ cells (both Tem and Tcm). The authors also still need to provide a control for their specific pentamer, while the irrelevant pentamer is a good control it doesn't control of issues with the specificity of the AH1 pentamer in your model. Please show staining in a non-tumor bearing mouse.

Response: The antigen-specific response of CD8⁺ T cells in Fig. 5j has been re-investigated by quantifying CD44⁺H-2Ld SPSYVYHQF-pentamer⁺ cells as suggested. The slow-release feature of LDH@aPD-1@CC patch could long-termly maintain the SLN a high proportion of CD8⁺ T cells with antigen-specific immune memory (CD44⁺H-2Ld SPSYVYHQF pentamer⁺, Fig. 5j), fighting against cancer resurgence.

Fig. 5j, Representative flow cytometric plots of CD44⁺H-2Ld SPSYVYHQF-pentamer⁺CD8⁺ T cells, gating on CD8⁺CD3⁺ cells in SLN.

For the control staining of the AH1 pentamer in non-tumor lymph node, we actually had done while demonstrating that the implanted patch has no obvious immune effect on the contralateral LNs to the SLNs. As shown in Supplementary Figure 12 (the previous Supplementary Figure 13), no H-2Ld SPSYVYHQF pentamer⁺CD8⁺ T cells were traced.

Supplementary Figure 12. Analyses of the LNs on the contralateral side of SLNs. **a**, Schematic illustrating the LN survey on the 4T1 BC mouse model after surgery, including no patch implantation (untreated, Group I) and patches with different therapeutic agents (CC, Group II; aPD-1@CC, Group III; LDH@CC, Group IV; and LDH@aPD-1@CC, Group V). **b**, Representative flow cytometric plots of H-2Ld SPSYVYHQF-pentamer⁺CD8⁺ T cells, gating on CD8⁺CD3⁺ cells, on Day 30 postsurgery. n = 5 mice per group.

Fig. 5j and its gating strategy in Supplementary Figure 10 have been re-prepared with the corresponding discussion in line 15-19, page 15, in the revised manuscript.

10. The added mechanistic data is all very interesting but a bit disconnected. I would recommend removing 3i and j as there is not link to any of the other data. Does anything in Figure 4 relate to the glycolysis findings in Figure 3? Please provide statistics for Figure 4d or remove claims of improved tumor cell killing at limiting dilutions. It is recommended that the key mechanistic insight be highlighted and the rest removed to focus the message.

Response: As suggested, we have removed Fig. 3i,j in the revised manuscript.

Fig. 4 reveals the enhancing mechanism of LDH adjuvant on the effector function of cytotoxic T lymphocytes (CTLs) via Mg²⁺/Fe³⁺-mediated shaping, which well explains the CTL-glycolytic upregulation by LDH in Fig. 3. Since that upregulated glycolysis is intimately linked to robust T-cell immunoresponses (*Science*, 2013, 342, 1242454), the engagement of aPD-1 and LDH in Fig. 3 would benefit the effector function of CTLs in SLN. It is well known that aPD-1 blocks PD-1 presented on CTLs for immune regulation (*Sci. Adv.*, 2020, 6, eabd2712; *Immunity*, 2022, 55, 512).

However, CTL-shaping mechanism by LDH adjuvant is uncertain and may be associated with the participation of Mg^{2+} and Fe^{3+} ions (*Cell*, 2022, 185, 585; *Trends Immunol.*, 2018, 39, 489; *Immunity*, 2013, 38, 225; *Nat. Immunol.*, 2004, 5, 818). Therefore, we comprehensively investigated the aforementioned speculation in Fig. 4.

Page 11 and line 22-23, page 12 have been revised to highlight the messages provided from Fig. 3 and Fig. 4, and the statistical analysis for Fig. 4d has been added in the revised manuscript.

Fig. 4d. A 2-h killing assay of CTLs with varying ratios of BALB/c CTLs to targets (B16F10 cancer cells). n = 3 per group. Data are presented as the mean \pm SD, and statistical significance was calculated via one-way ANOVA with Tukey's multiple comparisons.

11. Thank you for the images, but wound closure data should be quantified across all groups. This data is important with respect to the likelihood this kind of approach would ever be taken in patients. The rate of local recurrence should also be quantified. If no difference in rate of local recurrence I would recommend removing the immunophenotyping data at the wound site as it does not add in a meaningful way to explaining the therapeutic response driven by the patch.

Response: In fact, the rates of local tumor recurrence had been quantified. As shown in Fig. 6a-d, mice with surgery only (Group I) showed severe tumor recurrence (12/12), who was not affected by the implantation of biocompatible CC matrix (Group II, 12/12). The aPD-1@CC (Group III) and LDH@CC (Group IV) patch protected approximately half of the 12 mice from tumor recurrence for at least one month, and nevertheless, was not comparable with the effect of LDH@aPD-1@CC (Group V), in which 10 out of 12 mice were tumor-free.

Fig. 6 | Postsurgical 4T1-BC treatment of flex-patch. **a**, Schematic illustrating the therapeutic procedure in the 4T1 BC mouse model after surgery, including no patch implantation (untreated, Group I) and patches with different therapeutic agents (CC, Group II; aPD-1@CC, Group III; LDH@CC, Group IV; and LDH@aPD-1@CC, Group V). $n = 12$ mice per group. **b**, Representative bioluminescence images of the mice per group. **c,d**, Individual (**c**) and average (**d**) tumor growth kinetics per group. For (**d**), the curve ended when the first mouse in the corresponding group died. Statistical significance was calculated via one-way ANOVA with Tukey's multiple comparisons.

We agree that too many immunophenotyping data at the wound site did not support the therapeutic response driven by the patch in a meaningful way. Therefore, data analyzing macrophages of the wound have been removed. Moreover, the final section of “Flex-patch performs efficient postsurgical immunoadjuvant therapy” has been re-organized in the revised manuscript. Wound closure per group has been replenished in **Supplementary Figure 18d**.

Supplementary Figure 18d, Photographs of postsurgical wound per group.

As mentioned in the first review, the language remains overly complicated and should be significantly revised and simplified. It is very difficult to extract the key points and pieces of data from the text and figures.

Response: Thanks for the suggestion. We have simplified the manuscript. Moreover, by following the recommendation from the editor, the whole manuscript has been polished by the highly qualified native English speaking editors from the “American Journal Experts”.

REVIEWERS' COMMENTS

Reviewer #2 (Remarks to the Author):

My concerns and comments have been addressed satisfactorily.

Reviewer #4 (Remarks to the Author):

The authors have addressed all my concerns.